

# Narain CFTs from qudit stabilizer codes

Kohki Kawabata[1,2], Tatsuma Nishioka[2] and Takuya Okuda[3]

**1** Department of Physics, Faculty of Science, The University of Tokyo,
Bunkyo-Ku, Tokyo 113-0033, Japan
**2** Department of Physics, Osaka University, Machikaneyama-Cho 1-1,
Toyonaka 560-0043, Japan
**3** Graduate School of Arts and Sciences, The University of Tokyo,
Komaba, Meguro-ku, Tokyo 153-8902, Japan

## Abstract

We construct a discrete subset of Narain CFTs from quantum stabilizer codes with qudit (including qubit) systems whose dimension is a prime number. Our construction exploits three important relations. The first relation is between qudit stabilizer codes and classical codes. The second is between classical codes and Lorentzian lattices. The third is between Lorentzian lattices and Narain CFTs. In particular, we study qudit Calderbank-Shor-Steane (CSS) codes as a special class of qudit stabilizer codes and the ensembles of the Narain code CFTs constructed from CSS codes. We obtain exact results for the averaged partition functions over the ensembles and discuss their implications for holographic duality.

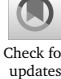

# 1   Introduction

The main goal of this paper is to construct a class of non-chiral conformal field theories (CFTs) from quantum error-correcting codes. It has been known for many years that a certain class of chiral CFTs can be constructed from classical error-correcting codes [1–3]. In recent years, an analogous construction for non-chiral CFTs has been developed in [4] based on a specific type of quantum error-correcting codes called qubit stabilizer codes, which results in a discrete subset of Narain CFTs named Narain code CFTs. We generalize this construction of Narain code CFTs to *qudit* stabilizer codes. The qudit system is a natural generalization of the qubit system to higher dimensions with $d$-level quantum states $|x\rangle$ ($x = 0, 1, \cdots, d-1$). Quantum error-correcting codes with qudit systems can be formulated in the same way [5] as in the binary case [6,7]. In this paper, we extend the construction from binary systems to $d$-ary systems for $d = p$ being a prime number.

We establish the relationship between qudit stabilizer codes, Lorentzian lattices, and Narain code CFTs in a similar manner to the binary case [4].[1] To this end, we leverage the following results in the literature:

- Some qudit stabilizer codes are associated with classical codes [8–10].

- Some Lorentzian lattices can be constructed from classical $p$-ary codes [11].

We combine these ingredients to construct Lorentzian lattices from qudit stabilizer codes (see figure 1). Then, we define a Narain code CFT by regarding each resulting Lorentzian lattice as the momentum lattice of the CFT. We show that the modular invariance of the Narain code CFT is guaranteed by certain conditions satisfied by the stabilizer code or equivalently by the classical code. The correspondences between qudit codes, Lorentzian lattices, and Narain CFTs are summarized in table 1.

In particular, our construction reveals a concrete relation among certain functions associated with codes, lattices, and CFTs. Let $\mathcal{C}$ be the classical code that specifies a qudit stabilizer

---

[1]While our construction closely follows the one in [4], there is a major difference between the binary and $p$-ary cases with odd-prime $p$. In our construction, equivalent qudit stabilizer codes do not necessarily yield the same Narain code CFT unless $p = 2$. See the comment in section 2.2.3 for more details.

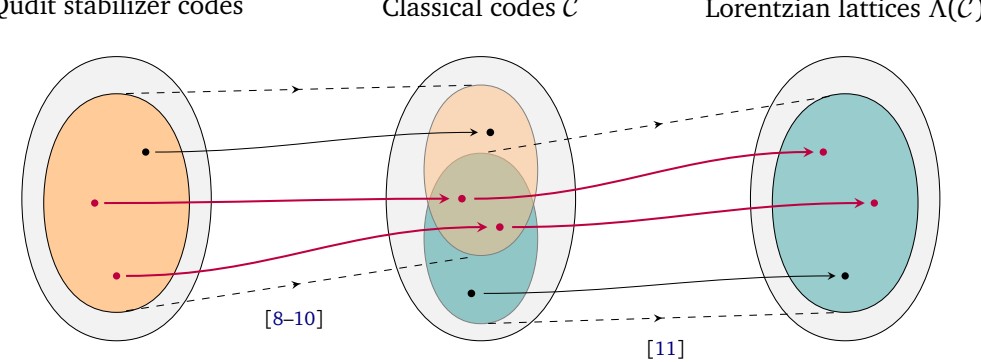

Figure 1: An illustration of our construction of Lorentzian lattices from qudit stabilizer codes. There is a class of classical codes associated with qudit stabilizer codes [8–10] (the light orange region in the middle ellipse). On the other hand, Lorentzian lattices can be built out of a certain class of classical codes [11] (the light green region in the middle ellipse). Focusing on the intersection of the two classes of classical codes allows us to construct a Lorentzian lattice from a qudit stabilizer code (the red arrows).

code. Then the CFT torus partition function $Z_{\mathcal{C}}(\tau, \bar{\tau})$, the lattice theta function $\Theta_{\widetilde{\Lambda}(\mathcal{C})}(\tau, \bar{\tau})$ for the lattice $\widetilde{\Lambda}(\mathcal{C})$, and the complete enumerator polynomial $W_{\mathcal{C}}(\{x_{ab}\})$ of $\mathcal{C}$ are related as

$$Z_{\mathcal{C}}(\tau, \bar{\tau}) = \frac{\Theta_{\widetilde{\Lambda}(\mathcal{C})}(\tau, \bar{\tau})}{|\eta(\tau)|^{2n}} = \frac{1}{|\eta(\tau)|^{2n}} W_{\mathcal{C}}(\{\psi_{ab}\}). \tag{1}$$

Here $\tau$ is the modulus of the torus, $\eta$ is the Dedekind eta function, and $\psi_{ab}$ are functions of $\tau$ and $\bar{\tau}$.[2] Thus the spectrum of the CFT can be read off from any of the three functions.

While our construction of Narain code CFTs is limited to a part of qudit stabilizer codes, it can be applied to an important class of quantum codes known as qudit Calderbank-Shor-Steane (CSS) codes. The CSS codes are quantum error-correcting codes defined by a pair $(C^{(1)}, C^{(2)})$ of classical codes [12, 13]. In this sense, CSS codes form a subset of quantum codes closely related to classical codes. Therefore, we can exploit the fundamental features of classical linear codes to analyze the CSS codes. Let us consider a CSS code defined by the pair $(C^{(1)}, C^{(2)}) = (C, C^{\perp})$ for a classical code $C$, where $C^{\perp}$ is the dual code of $C$. Then, a Narain code CFT associated with the CSS code can be constructed, whose partition function is uniquely determined by the complete joint weight enumerator $\mathcal{W}_{\underline{C}}(\{x_{ab}\})$ of $C$ and $C^{\perp}$ introduced in [14]:

$$Z_{C,C^{\perp}}^{(\text{CSS})}(\tau, \bar{\tau}) = \frac{1}{|\eta(\tau)|^{2n}} \mathcal{W}_{\underline{C}}(\{\psi_{ab}\}), \tag{2}$$

where $\underline{C} = C \times C^{\perp}$. The complete joint weight enumerator was originally introduced in the study of classical codes. We will also give a few simple examples for Narain code CFTs based on CSS codes and exemplify our construction in more detail in section 4.3.

To investigate the universal aspects of the Narain code CFTs we construct, we consider the partition functions averaged over a class of CSS codes. Recently, ensemble averaging of Narain CFTs has attracted much attention with a view to seeking a holographic duality [15, 16] (see [17–29] for related works). In this paper, we focus on CSS codes $\mathcal{C}$ given by

---

[2]Explicitly, $\psi_{ab}(\tau, \bar{\tau})$ are defined in (90) and can be rewritten as (95).

$(C^{(1)}, C^{(2)}) = (C, C)$ and average over self-dual classical codes $C$. The partition function of the Narain code CFT based on a single such CSS code $\mathcal{C}$ turns out to be the genus-2 complete enumerator polynomial $W_{2,C}(\{x_{ab}\})$ of the self-dual code $C$:

$$Z_{C,C}^{(\text{CSS})}(\tau, \bar{\tau}) = \frac{1}{|\eta(\tau)|^{2n}} W_{2,C}(\{\psi_{ab}\}). \tag{3}$$

Then, the average over self-dual codes takes the form

$$\overline{Z}_{n,p}^{(\text{CSS})}(\tau, \bar{\tau}) := \frac{1}{|\mathcal{M}_{n,p}|} \sum_{C \in \mathcal{M}_{n,p}} Z_{C,C}^{(\text{CSS})}(\tau, \bar{\tau}) = \frac{1}{|\eta(\tau)|^{2n}} \frac{1}{|\mathcal{M}_{n,p}|} \sum_{C \in \mathcal{M}_{n,p}} W_{2,C}(\{\psi_{ab}\}), \tag{4}$$

where $\mathcal{M}_{n,p}$ is the set of all classical $p$-ary self-dual codes of length $n$. Hence, our problem amounts to calculating the average of the enumerator polynomials over self-dual codes $C$.

While we are mainly concerned with the genus-2 case, we address the more general problems of calculating the average of the genus-$g$ complete enumerator polynomial $W_{g,C}$ over the set $\mathcal{M}_{n,p}$,

$$E_{n,p}^{(g)}(\{x_v\}) = \frac{1}{|\mathcal{M}_{n,p}|} \sum_{C \in \mathcal{M}_{n,p}} W_{g,C}(\{x_v\}). \tag{5}$$

The formula for the average of the genus-$g$ complete enumerator polynomial over doubly-even self-dual codes was given in [30, 31]. To our best knowledge, however, the averaged genus-$g$ complete enumerator polynomial for self-dual codes has not been derived yet. The properties of classical self-dual codes allow us to explicitly write down the formula for $p = 2$ in Theorem 5.2 and for odd prime $p$ in Theorem 5.4. Therefore, focusing on the genus-2 case, we obtain the exact averaged partition functions (3) of the CSS codes. We find that the averaged partition function reproduces an averaged partition function conjectured in [28] for a similar but different ensemble of codes in the large central charge limit. We will discuss the implications of the averaged Narain code CFTs for holographic duality in section 6 along the line of [15, 16].

The organization of this paper is as follows. In section 2, we review the qudit stabilizer formalism and in particular the symplectic representation that we use. After introducing these elements, we concretely illustrate qudit codes by giving some examples of CSS codes. In section 3, we examine the conditions for a qudit stabilizer code to yield an even self-dual lattice and point out that a class of CSS codes satisfies the conditions automatically. In section 4, the resulting Lorentzian even self-dual lattices are lifted to Narain code CFTs, and the dictionary between codes, lattices, and CFTs is given. In section 5, we consider the averaged theory of Narain code CFTs. We give the general formula for the average of the higher-genus weight enumerators, which reduces to the averaged partition function for $g = 2$. We point out that our result exactly agrees with the conjectural partition function of the averaged theory associated with error-correcting codes in [28]. Section 6 concludes with discussions and future directions. Appendix A lists our notations used throughout this paper. In appendix B, we give details for a saddle point computation in section 5.

## 2 Qudit stabilizer codes

In this section, we will review quantum error correction on qudit systems, which is the generalization of a qubit to higher dimensions following [5, 32, 33]. We illustrate quantum error-correcting codes focusing on stabilizer codes in section 2.2. In section 2.3, we introduce CSS codes, a class of stabilizer codes constructed from a pair of classical codes. We will see later that CSS codes are compatible with our construction of Narain CFTs.

## 2.1 Qudit system

We consider a $d$-level quantum system called a qudit system (refer to Appendix A.1 in [34] and section 2 in [35]). For simplicity, we set the number of states with the qudit system as a prime $d = p$. Then, a qudit state takes over a finite field $\mathbb{F}_p = \mathbb{Z}/p\,\mathbb{Z}$. An orthonormal basis on a qudit system $H_p$ is given by $\{|x\rangle\}_{x=0}^{x=p-1}$. The elementary actions on the Hilbert space $H_p$ are given by

$$X_p |x\rangle = |x+1\rangle, \qquad Z_p |x\rangle = \omega_p^x |x\rangle, \tag{6}$$

where $\omega_p = e^{2\pi i/p}$ and $x \in \mathbb{F}_p$: $x \sim x + p$. These operators are called the qudit Pauli $X$ and $Z$ operator [5]. The qudit Pauli operators are represented by

$$X_p = \sum_{x=0}^{p-1} |x+1\rangle \langle x|, \qquad Z_p = \sum_{x=0}^{p-1} \omega_p^x |x\rangle \langle x|. \tag{7}$$

Therefore, we have the following commutation relation:

$$Z_p X_p = \omega_p X_p Z_p. \tag{8}$$

For example, these operators become Pauli $X$ and Pauli $Z$ when the system is a qubit ($p = 2$). In the case of a qutrit ($p = 3$), these operators are $3 \times 3$ matrices.

$$X_3 = \begin{bmatrix} 0 & 0 & 1 \\ 1 & 0 & 0 \\ 0 & 1 & 0 \end{bmatrix}, \qquad Z_3 = \begin{bmatrix} 1 & 0 & 0 \\ 0 & \omega_3 & 0 \\ 0 & 0 & \omega_3^2 \end{bmatrix}, \tag{9}$$

where $\omega_3 = e^{2\pi i/3}$. We define generalized Pauli operators that act on a qudit system as

$$g(\alpha, \beta) = \omega^\kappa X_p^\alpha Z_p^\beta = \omega^\kappa \sum_{x=0}^{p-1} \omega_p^{x\beta} |x+\alpha\rangle \langle x|, \tag{10}$$

where $\alpha, \beta \in \mathbb{F}_p = \{0, 1, \cdots, p-1\}$. We suppress the dependence on $\kappa$ in $g(\alpha, \beta)$ because it plays no role for our construction of Narain CFTs. The global phase factor is given by

$$\omega^\kappa = \begin{cases} i^\kappa, & \text{if } p = 2, \\ \omega_p^\kappa, & \text{if } p \text{ odd prime}, \end{cases} \tag{11}$$

where $\kappa \in \{0, 1, 2, 3\}$ for $p = 2$ and $\kappa \in \mathbb{F}_p$ for an odd prime. This ensures that there exists a choice of $\kappa$ in operators $g(\alpha, \beta)$ such that $g(\alpha, \beta)^p = 1$ for any $\alpha, \beta \in \mathbb{F}_p$ [5]. There are $p^2$ operators up to phases, which act on a qudit system in an analogous way to four operators $\{I, X, Y, Z\}$ in a qubit system. The commutation relations are

$$g(\alpha, \beta)\,g(\alpha', \beta') = \omega_p^{-\alpha\beta' + \beta\alpha'}\,g(\alpha', \beta')\,g(\alpha, \beta). \tag{12}$$

Then two operators commute if and only if $\alpha\beta' - \beta\alpha' = 0 \bmod p$.

We can easily generalize this representation to the $n$-qudit system. An orthonormal basis in the $n$-qudit system is the $n$-fold tensor products of $\{|x\rangle\}_{x=0}^{x=p-1}$. The $p^{2n}$ operators that act on the $n$-qudit system are given by

$$g(\alpha, \beta) = g(\alpha_1, \beta_1) \otimes \cdots \otimes g(\alpha_n, \beta_n) = \omega^\kappa X_p^{\alpha_1} Z_p^{\beta_1} \otimes \cdots \otimes X_p^{\alpha_n} Z_p^{\beta_n}, \tag{13}$$

where $\alpha = (\alpha_1, \cdots, \alpha_n) \in \mathbb{F}_p^n$, $\beta = (\beta_1, \cdots, \beta_n) \in \mathbb{F}_p^n$ and the global phase is given by (11). We call the group that acts on the $n$-qudit system the $n$-qudit Pauli group $\mathcal{P}_n^{(p)}$. For odd prime $p$, the elements of $\mathcal{P}_n^{(p)}$ have eigenvalues $\omega_p^i$ for $i = 0, 1, \cdots, p-1$. For the case of qubits ($p = 2$), the group $\mathcal{P}_n^{(2)}$ consists of all $n$-fold tensor products of the Pauli matrices multiplied by $\pm 1$ or $\pm i$. These elements have eigenvalues of either $\pm 1$ or $\pm i$. The commutation relations are given by

$$g(\alpha, \beta)\, g(\alpha', \beta') = \omega_p^{-\alpha \cdot \beta' + \beta \cdot \alpha'}\, g(\alpha', \beta')\, g(\alpha, \beta), \tag{14}$$

where we introduce the dot product

$$\alpha \cdot \beta = (\alpha_1, \cdots, \alpha_n) \cdot (\beta_1, \cdots, \beta_n) = \sum_{i=1}^{n} \alpha_i \beta_i, \tag{15}$$

where arithmetic is performed in $\mathbb{F}_p$ (modulo $p$). It may be useful to introduce the following symplectic product [8]:

$$\langle (\alpha, \beta), (\alpha', \beta') \rangle = \alpha \cdot \beta' - \beta \cdot \alpha'. \tag{16}$$

Then, the commutation relations imply that a pair of operators $g(\alpha, \beta)$, $g(\alpha', \beta')$ commute each other if and only if the symplectic product vanishes: $\langle (\alpha, \beta), (\alpha', \beta') \rangle = 0 \mod p$.

## 2.2 Stabilizer codes

Error-correcting codes were invented to communicate with others via a noisy channel. We send an original message encoded as an appropriate signal to be able to correct some noise. In quantum error-correcting codes, we send a quantum state as an encoded signal. For specifying an encoded quantum state, some group theoretic methods are useful. Such a class of quantum codes is called stabilizer codes.

### 2.2.1 Stabilizer formalism

In order to understand stabilizer codes, we must develop a stabilizer formalism. The stabilizer formalism is convenient for representing the state vector compactly in a clever use of group theory. Stabilizer codes were originally considered for qubits by Gottesman [6]. After that, the notion of stabilizer codes was generalized to qudits in [36–38].

Suppose that $S$ is an abelian subgroup of $\mathcal{P}_n^{(p)}$, called the stabilizer group. The set of valid codewords forms a subspace of the full $n$ qudit Hilbert space, the code subspace of the quantum code. For a stabilizer group $S$, a code subspace $V_S$ is composed of states that are fixed by all elements of $S$: for $|\psi\rangle \in V_S$,

$$g |\psi\rangle = |\psi\rangle, \qquad g \in S. \tag{17}$$

The projector on the code subspace $V_S$ is given by

$$P_S = \frac{1}{|S|} \sum_{g \in S} g. \tag{18}$$

Actually, this operator satisfies $P_S^2 = P_S$ due to the group structure of the stabilizer group $S$.

A qubit stabilizer code with a nontrivial code subspace $V_S$ must have an abelian stabilizer group $S$ that does not contain $\pm i I$ [39]. A similar proposition holds for odd prime $p$.

**Proposition 2.1**

Let $S$ be a subgroup of the $n$-qudit Pauli group $\mathcal{P}_n^{(p)}$ for odd prime $p$. The group $S$ is an abelian group which does not contain $\omega_p^i I$ ($i = 1, 2, \cdots, p-1$) if the stabilizer code has a nontrivial code subspace $V_S$.

*Proof.* In the following, we prove that a code subspace becomes trivial assuming that the stabilizer group is non-abelian or has a nontrivial multiple of the identity.

Firstly, let us consider the case when a non-abelian subgroup $S$ of $\mathcal{P}_n^{(p)}$ stabilizes a code subspace $V_S$. Suppose that $M, N \in S$ stabilize a state $|\psi\rangle \in V_S$. Then $|\psi\rangle = MN|\psi\rangle = \omega_p^i NM|\psi\rangle = \omega_p^i|\psi\rangle$ for some $i \in \{1, 2, \cdots, p-1\}$. This implies the encoded state $|\psi\rangle$ is trivial: $|\psi\rangle = 0$. Next, assume that an abelian stabilizer group $S$ contains a nontrivial multiple of identity. Then we have $\omega_p^i I \in S$ where $i = 1, 2, \cdots, p-1$, so $V_S \ni |\psi\rangle = \omega_p^i I|\psi\rangle = \omega_p^i|\psi\rangle$. We conclude that $|\psi\rangle = 0$.

$\square$

The stabilizer group can be characterized by $n - k$ independent generators $g_1, \cdots, g_{n-k}$. More concretely, the stabilizer group is generated by $g(\alpha^{(1)}, \beta^{(1)}), \cdots, g(\alpha^{(n-k)}, \beta^{(n-k)})$ on an $n$ qudit system where $(\alpha^{(i)}, \beta^{(i)}) \in \mathbb{F}_p^n \times \mathbb{F}_p^n$ specifies the generators of the stabilizer group. The stabilizer generator $g_i \in S$ divides the entire $p^n$-dimensional Hilbert space into $p$ subspaces of equal dimension by its eigenvalue. Since there are $(n-k)$ stabilizer generators, $V_S$ is a $p^k$-dimensional vector space. In this case, a stabilizer code is called $[[n, k]]_p$ code.

Stabilizer groups $S$ stabilize states in the code subspace $V_S$. On the other hand, there are operators that change states from the code subspace into other states in the code subspace. These operators are called *logical operators*. Logical operators do not map the encoded state $|\psi\rangle \in V_S$ into a non-code subspace. It follows that stabilizer operators and logical operators commute. Let us illustrate this fact. Suppose that an operator $E_L$ does not commute with a stabilizer operator $g \in S$. Then we have

$$g E_L |\psi\rangle = \omega^\kappa E_L g |\psi\rangle = \omega^\kappa E_L |\psi\rangle, \qquad |\psi\rangle \in V_S, \tag{19}$$

where the phase factor is nontrivial ($\kappa \neq 0$). The stabilizer operator $g \in S$ does not stabilize the state $E_L |\psi\rangle$ and then $E_L |\psi\rangle \notin V_S$. This implies that if an operator $E_L$ does not commute with a stabilizer operator, then the action of $E_L$ on a coding state $|\psi\rangle \in V_S$ put into a non-code subspace: $E_L |\psi\rangle \notin V_S$. Therefore, to stay in a code subspace under the action of an operator $E_L$, this operator $E_L$ has to commute with a stabilizer group $S$.

We can write logical operators as the $n$-fold tensor products (13). Since the number of all operators that act on $k$ qudits is $p^{2k}$ up to global phase factors, the same number of logical operators act on the encoded subspace. Then, we have $2k$ generators of such transformations. There are $p^{2n}$ operators that act on the $n$-qudits system in all. $p^{n-k}$ of them are stabilizer operators, and $p^{2k}$ of them are logical operators. Other $p^{n-k}$ operators that anticommute with the stabilizer group are called *error operators*.

These operators can be recast in a more group theoretically sophisticated manner. Let us pick up an abelian subgroup $S$ of the $n$-qudit Pauli group $\mathcal{P}_n^{(p)}$. For each stabilizer group $S$, we introduce the normalizer (or centralizer) $N(S)$ of $S$ in $\mathcal{P}_n^{(p)}$, i.e., the subgroup of $\mathcal{P}_n^{(p)}$ containing all elements that commutes with every element of $S$. Then, logical operators are defined as elements of $N(S) \backslash S$. Also, error operators that anticommute with each element of $S$ are given by elements of $\mathcal{P}_n^{(p)} \backslash N(S)$. Note that the set of logical operators and the set of error operators cannot have the group structure since the identity is always in the stabilizer group $S$.

### 2.2.2 Symplectic representation

In the above, we have described a stabilizer code using an operator formalism. We can encode a stabilizer group $S$ into an $(n-k) \times 2n$ check matrix [8]:

$$
\mathsf{H} = \left[\begin{array}{ccc|ccc}
 & \alpha^{(1)} & & & \beta^{(1)} & \\
 & \alpha^{(2)} & & & \beta^{(2)} & \\
 & \vdots & & & \vdots & \\
 & \alpha^{(n-k)} & & & \beta^{(n-k)} &
\end{array}\right],
\tag{20}
$$

where $(\alpha^{(i)}, \beta^{(i)}) \in \mathbb{F}_p^n \times \mathbb{F}_p^n$ characterizes the generators of the stabilizer group $S$. In general, a stabilizer generator has a phase factor $\omega^\kappa$ that is not considered in the above check matrix:

$$
g(\alpha, \beta) = \omega^\kappa X^{\alpha_1} Z^{\beta_1} \otimes \cdots \otimes X^{\alpha_n} Z^{\beta_n},
\tag{21}
$$

where $\alpha = (\alpha_1, \cdots, \alpha_n)$ and $\beta = (\beta_1, \cdots, \beta_n)$. By code equivalence we can set $\omega^\kappa$ to 1 for odd prime $p$ and to $\mathrm{i}^{\alpha \cdot \beta}$ for $p = 2$.[3]

A stabilizer group $S$ is mapped to a check matrix $\mathsf{H}$. The commutation relation in the stabilizer group is also encoded into a symplectic product on the vector space spanned by the check matrix. We define a $2n \times 2n$ matrix $\mathsf{W}$ as

$$
\mathsf{W} = \left[\begin{array}{cc}
0 & I_n \\
-I_n & 0
\end{array}\right],
\tag{22}
$$

where the $I_n$ in the off-diagonals is an $n \times n$ identity matrix. Elements $g(\alpha, \beta)$ and $g(\alpha', \beta')$ commute if and only if $(\alpha, \beta) \mathsf{W} (\alpha', \beta')^T = \langle (\alpha, \beta), (\alpha', \beta') \rangle = 0$. Then the abelian structure of a stabilizer group reduces to the following condition:

$$
\mathsf{H} \mathsf{W} \mathsf{H}^T = 0 \qquad \mod p,
\tag{23}
$$

where 0 on the right-hand side denotes a $(n-k) \times (n-k)$ matrix.

We introduce the generator matrix $\mathsf{G}$ over $\mathbb{F}_p$ such that

$$
\mathsf{H} \mathsf{W} \mathsf{G}^T = 0 \qquad \mod p,
\tag{24}
$$

where $\mathsf{G}$ is a $(n+k) \times 2n$ matrix with $\mathrm{rank}(\mathsf{G}) = n+k$ and its component is given by

$$
\mathsf{G} = \left[\begin{array}{ccc|ccc}
 & \alpha^{(1)} & & & \beta^{(1)} & \\
 & \vdots & & & \vdots & \\
 & \alpha^{(n+k)} & & & \beta^{(n+k)} &
\end{array}\right].
\tag{25}
$$

This implies that the operators generated by rows of the generator matrix commute with the stabilizer group. The generator matrix $\mathsf{G}$ generates the normalizer $N(S)$ of the stabilizer group $S$ in $\mathcal{P}_n^{(p)}$, which consists of stabilizer operators and logical operators. We can choose the generator matrix $\mathsf{G}$ such that the first $(n-k)$ rows and the remaining $2k$ rows generate stabilizer operators and the set of logical operators, respectively.

---

[3]This statement follows from Proposition 2.2.

### 2.2.3 Code equivalence

There is a subgroup of unitary transformations that do not change the form (13) of the stabilizer generators. The group with this property is called the Clifford group. The Clifford group is characterized by the property that it leaves the $n$-qudit Pauli group $\mathcal{P}_n^{(p)}$ invariant under conjugation. Hence, it is a normalizer of the qudits Pauli group: $N\left(\mathcal{P}_n^{(p)}\right)$ in the unitary group $U(p^n)$. The Clifford group generates equivalence classes of the stabilizer codes by conjugation. The stabilizer codes in the same equivalence class are called equivalent.

For the case with qubits ($p = 2$), the Clifford group is generated by the Hadamard transformation: $X \to Z$, $Z \to X$ and the phase gate: $X \to PXP^{-1}$, $Z \to Z$ where $P = \mathrm{diag}(1, i)$, and the CNOT gate:

$$
\begin{aligned}
X \otimes I &\to X \otimes X, & I \otimes X &\to I \otimes X, \\
Z \otimes I &\to Z \otimes I, & I \otimes Z &\to Z \otimes Z.
\end{aligned}
\tag{26}
$$

For qudits where $p$ is an odd prime, there are the following transformations in the Clifford group [5], called the discrete Fourier transformation: $X_p \to Z_p$, $Z_p \to X_p^{-1}$ and the phase gate: $X_p \to X_p Z_p$, $Z_p \to Z_p$, and the SUM gate:

$$
\begin{aligned}
X_p \otimes I &\to X_p \otimes X_p, & I \otimes X_p &\to I \otimes X_p, \\
Z_p \otimes I &\to Z_p \otimes I, & I \otimes Z_p &\to Z_p^{-1} \otimes Z_p.
\end{aligned}
\tag{27}
$$

Furthermore, we need the $S$ gate to generate the Clifford group: $X_p \to X_p^a$, $Z_p \to Z_p^b$ where $ab = 1 \bmod p$. These four operators generate the Clifford group $N\left(P_n^{(p)}\right)$. Then a stabilizer code is equivalent to another code obtained by the conjugation generated by these operators.

Related to the code equivalence, we can show the following proposition. This statement ensures the existence of an equivalent stabilizer code with trivial phases.

**Proposition 2.2**
Suppose that the stabilizer generators be $g_i$ where $i = 1, 2, \cdots, n-k$. For fixed $i$, there exists $g \in \mathcal{P}_n^{(p)}$ such that $g\, g_i\, g^{-1} = \omega_p^{\kappa} g_i$ for $\kappa \in \{1, 2, \cdots, p-1\}$ and $g\, g_j\, g^{-1} = g_j$ for $j \neq i$.

*Proof.* Suppose that a check matrix $\mathsf{H}$ of a stabilizer group $S$ is of the form (20) where the rows are linearly independent. Then there exists a $2n$-dimensional row vector $x = (\alpha, \beta) \in \mathbb{F}_p^n \times \mathbb{F}_p^n$ which satisfies

$$
\mathsf{H}\,\mathsf{W}\,x^T = e_i,
\tag{28}
$$

where $e_i$ is the $(n+k)$-dimensional column vector with 1 at the $i$-th position and 0s elsewhere. Let $\sigma \in \mathcal{P}_n^{(p)}$ be an operator such that

$$
\sigma = g(\alpha, \beta).
\tag{29}
$$

Let $g_i \in S$ be a generator of the stabilizer group encoded in the $i$-th row of the check matrix $\mathsf{H}$. Then we have the following relation from (28): $g_i \sigma = \omega_p^{-1} \sigma g_i$, and $g_j \sigma = \sigma g_j$ where $j \neq i$. This implies that $\sigma = g(\alpha, \beta) \in \mathcal{P}_n^{(p)}$ acts as $\sigma g_i \sigma^{-1} = \omega_p g_i$ and $\sigma g_j \sigma^{-1} = g_j$. Hence, we obtain the result $g\, g_i\, g^{-1} = \omega_p^{\kappa} g_i$ and $g\, g_j\, g^{-1} = g_j$ where $g = \sigma^{\kappa}$. $\qquad\square$

Since the Clifford group is the normalizer of the Pauli group $\mathcal{P}_n^{(p)}$, it contains the Pauli group $\mathcal{P}_n^{(p)}$. Proposition 2.2 states that there exists an equivalent stabilizer code that is the same as the original code except for phase factors. Therefore, it allows us to remove the phase factors in front of the stabilizer generators by an appropriate equivalent transformation.

Associated with the equivalence of quantum codes, we make a comment on our construction of Lorentzian lattices from qudit stabilizer codes illustrated in section 3.

**Comment:** Suppose that a qudit stabilizer code has a check matrix H. When constructing a Lorentzian lattice, we will introduce a Lorentzian metric $\eta$ into a vector space generated by the matrix H by hand. In the binary case ($p = 2$), the symplectic structure W inherited from quantum codes also undertakes the role of the Lorentzian metric as a result of the modulo-two operation: $W = \eta \bmod 2$. However, they do not match and must be defined separately for an odd prime $p$. Therefore, the Lorentzian metric and symplectic structure impose independent conditions when constructing Narain CFTs for an odd prime $p$. The Clifford group preserves only the symplectic structure and changes the Lorentzian metric. Thus, even if a quantum code satisfies the conditions for the construction of Narain CFTs, it is not guaranteed that an equivalent quantum code after the action of the Clifford group meets the same conditions when $p$ is an odd prime.

## 2.3 CSS codes

There is a class of stabilizer codes that can be constructed from a pair of classical codes. These codes are called CSS codes [12, 13]. CSS codes give nontrivial examples for our construction of Narain CFTs. To introduce CSS codes, we first illustrate classical linear codes briefly (see [40–45] for more details).

Let us define a $p$-ary classical linear code that encodes a $k$-bit message into an $n$-bit signal. A classical linear code $C$ has the generator matrix $G_C$ and the parity check matrix $H_C$ that satisfies

$$G_C H_C^T = 0 \qquad \bmod p \,, \tag{30}$$

where $G_C$ and $H_C$ are a $k \times n$ matrix of rank $k$ and an $(n-k) \times n$ matrix of rank $n-k$, respectively. The codewords are generated by the generator matrix $G_C$ as follows:

$$c = x \, G_C \,, \tag{31}$$

where $x \in \mathbb{F}_p^k$ is a $k$-dimensional row vector. These codewords determine the code subspace

$$C = \left\{ c \in \mathbb{F}_p^n \mid c = x \, G_C \,, \;\; x \in \mathbb{F}_p^k \right\} \subset \mathbb{F}_p^n \,. \tag{32}$$

For all codewords $c \in C$, the parity check matrix $H_C$ satisfies $c \, H_C^T = 0 \bmod p$ due to the condition (30). Then, the parity check matrix gives an alternative definition of the code subspace $C$

$$C = \left\{ c \in \mathbb{F}_p^n \mid c \, H_C^T = 0 \;\; \bmod p \right\} \,. \tag{33}$$

To characterize the error-correcting property of a linear code, let us introduce the distance in the vector space $\mathbb{F}_p^n$. The Hamming distance $d(c, c')$ between vectors $c, c' \in \mathbb{F}_p^n$ is given by the number of nonzero components of the vector $c - c' \in \mathbb{F}_p^n$. For a linear code, the Hamming weight is also useful. The Hamming weight $\mathrm{wt}(c)$ of a vector $c \in \mathbb{F}_p^n$ is defined as the number of nonzero components of the vector $c$. For example, the Hamming weight of the vector $c = (0, 0, 4, 3) \in \mathbb{F}_5^4$ is $\mathrm{wt}(c) = 2$. Using the Hamming distance or weight, we define the minimum distance of a linear code. The minimum distance $d(C)$ of a linear code $C$ is given by the minimum nonzero Hamming distance for any pair of codewords:

$$d(C) = \min_{c, c' \in C, c \neq c'} d(c, c') = \min_{c \in C, c \neq 0} \mathrm{wt}(c) \,, \tag{34}$$

where we use the fact that for a linear code $C$, $c - c' \in C$ if $c$ and $c'$ are codewords. A linear code with the minimum distance $d$ can correct up to $\lfloor (d-1)/2 \rfloor$ bit errors, so the minimum distance

captures the characteristics of the error-correcting property well. We call a $p$-ary linear code that encodes $k$ bits into $n$ bits with the minimum distance $d$ as an $[n, k, d]_p$ code. Often the minimum distance is omitted and simply referred to as an $[n, k]$ code.

A key ingredient in the CSS codes is the dual construction of classical codes. The dual code $C^\perp$ for a code $C$ is defined by

$$C^\perp = \{c' \in \mathbb{F}_p^n \,|\, c \cdot c' = 0 \ \mathrm{mod}\ p\,,\ c \in C\}\,. \tag{35}$$

We call a code *self-orthogonal* if $C \subseteq C^\perp$, and *self-dual* if $C = C^\perp$.

Suppose $C$ is a $p$-ary classical linear code with a $k \times n$ generator matrix $G_C$ and an $(n-k) \times n$ parity check matrix $H_C$. We assume the Euclidean metric $c \cdot c' = \sum_{i=1}^n c_i c_i'$. Then the dual code $C^\perp$ is the code with an $(n-k) \times n$ generator matrix $H_C$ and a $k \times n$ parity check matrix $G_C$. The codewords $c \in C$ are generated by the generator $G$: $c = x\, G_C$ where $x \in \mathbb{F}_p^k$ is a $k$-dimensional row vector. Also, the codewords $c' \in C^\perp$ are given by $c' = y\, H_C$ where $y \in \mathbb{F}_p^{n-k}$ is an $(n-k)$-dimensional row vector. The inner product of these vectors is $c \cdot c' = x\, G_C H_C^T y^T = 0$ mod $p$ from the relation (30).

Suppose that $C_X$ and $C_Z$ are $[n, k_X]_p$ and $[n, k_Z]_p$ linear codes with the generator matrices $G_X, G_Z$ and the parity check matrices $H_X, H_Z$, respectively. Also, we assume the following condition:

$$C_X^\perp \subseteq C_Z\,. \tag{36}$$

This condition implies that the dual code of $C_X$ is a subspace of the other code $C_Z$, so all codewords generated by $H_X$ are contained in the code subspace $C_Z$. Then we reach

$$H_X H_Z^T = 0 \qquad \mathrm{mod}\ p\,. \tag{37}$$

In this case, the CSS code can be defined by the following check matrix:

$$\mathsf{H}_{(C_X, C_Z)} = \left[ \begin{array}{c|c} H_X & 0 \\ 0 & H_Z \end{array} \right]\,, \tag{38}$$

where the block $H_X$ ($H_Z$) represents the parity check matrix of the classical linear code $C_X$ ($C_Z$). To see that this construction defines a stabilizer code, let us examine if the check matrix satisfies the commutativity condition (23): $\mathsf{H}_{(C_X, C_Z)} \mathsf{W} \mathsf{H}_{(C_X, C_Z)}^T = 0$ (mod $p$). Now we have the relation (37), then

$$\mathsf{H}_{(C_X, C_Z)} \mathsf{W} \mathsf{H}_{(C_X, C_Z)}^T = \left[ \begin{array}{cc} 0 & H_X H_Z^T \\ -H_Z H_X^T & 0 \end{array} \right] = 0 \qquad \mathrm{mod}\ p\,. \tag{39}$$

Therefore, the CSS code with the check matrix (38) is a subclass of the stabilizer code. The resulting qudit code is $[[n, k_X + k_Z - n]]_p$ type.

For self-dual codes $C$, we can choose the generator matrix as $G_C = H_C$. From (30), we have $H_C H_C^T = 0$ and this implies that if we choose $C_X = C_Z = C$, the commutativity condition (37) holds automatically. We can always construct the CSS code by setting $C_X = C_Z = C$ with a self-dual code $C$. In this case, we obtain a quantum $[[n, 0]]_p$ code since the classical self-dual codes satisfy $k = n/2$.

An example of the CSS codes is the three-qutrit code. Consider a classical ternary code $C$ with the generator matrix $G_3$ and the parity check matrix $H_3$:

$$G_3 = \left[ \begin{array}{ccc} 1 & 1 & 1 \\ 0 & 1 & 2 \end{array} \right]\,, \qquad H_3 = \left[ \begin{array}{ccc} 1 & 1 & 1 \end{array} \right]\,. \tag{40}$$

This code satisfies $C^\perp \subseteq C$ but is not self-dual $C \neq C^\perp$. We set $C_X = C_Z = C$. The commutativity condition (37) is satisfied for $H_X = H_Z = H_3$. Then the CSS code is given by

$$H_{(C,C)} = \begin{bmatrix} 1 & 1 & 1 & 0 & 0 & 0 \\ 0 & 0 & 0 & 1 & 1 & 1 \end{bmatrix}. \tag{41}$$

This check matrix gives us the stabilizer generators $g_1 = X \otimes X \otimes X$ and $g_2 = Z \otimes Z \otimes Z$. The stabilizer group generated by these operators stabilizes the following quantum codewords:[4]

$$\begin{aligned} |\bar{0}\rangle &= \frac{1}{\sqrt{3}} \left( |000\rangle + |111\rangle + |222\rangle \right), \\ |\bar{1}\rangle &= \frac{1}{\sqrt{3}} \left( |012\rangle + |120\rangle + |201\rangle \right), \\ |\bar{2}\rangle &= \frac{1}{\sqrt{3}} \left( |021\rangle + |102\rangle + |210\rangle \right). \end{aligned} \tag{42}$$

We give one more example of the CSS codes. There is a self-dual code $C$ over $\mathbb{F}_5$ of length $n = 2$. This classical code is given by the following generator matrix:

$$G_5 = \begin{bmatrix} 1 & 2 \end{bmatrix}. \tag{43}$$

Since the above code is self-dual, we can choose the parity matrix $H_5 = G_5$. For the same reason, we can choose $H_X = H_Z = H_5$ while satisfying the commutativity condition. Then the corresponding CSS code is

$$H_{(C,C)} = \begin{bmatrix} 1 & 2 & 0 & 0 \\ 0 & 0 & 1 & 2 \end{bmatrix}. \tag{44}$$

The stabilizer generators generated by the above check matrix are $g_1 = X \otimes X^2$ and $g_2 = Z \otimes Z^2$. These operators generate the stabilizer group $S$ and stabilize the following encoded state:

$$|\psi\rangle = \frac{1}{\sqrt{5}} \left( |00\rangle + |12\rangle + |24\rangle + |31\rangle + |43\rangle \right). \tag{45}$$

# 3 Construction of Lorentzian even self-dual lattices

Classical binary codes are known to give rise to Euclidean lattices and chiral CFTs [3]. In the previous section, we have described qudit stabilizer codes. In what follows, we will give an explicit construction of Lorentzian lattices from qudit stabilizer codes. In particular, we will illustrate that our construction works for the CSS codes. This is the generalization of the work [4], where the authors focus on the binary quantum stabilizer codes.

## 3.1 Lorentzian lattices via Construction A

A stabilizer code is defined by an abelian subgroup of the Pauli group $\mathcal{P}_n^{(p)}$, and the generators of each code are given by the rows of the check matrix (20). We define a classical code generated by the check matrix of a stabilizer code. We construct the Lorentzian lattice from the classical code and connect the property of a classical code and a lattice. In the following, we focus on an $[[n, 0]]_p$ qudit stabilizer code where the check matrix is an $n \times 2n$ matrix.

---

[4]The three-qutrit code can be seen as the simplest model of holography [46]. The relation between CSS codes and holography is also discussed in [47].

Suppose that a stabilizer code has the $n \times 2n$ check matrix

$$
H = \begin{bmatrix} \alpha^{(1)} & \beta^{(1)} \\ \alpha^{(2)} & \beta^{(2)} \\ \vdots & \vdots \\ \alpha^{(n)} & \beta^{(n)} \end{bmatrix}, \tag{46}
$$

where the rows are linearly independent since each row corresponds to an independent generator of the stabilizer group $S$. Then the rank of the check matrix is $\mathrm{rank}(H) = n$.

Consider a classical code generated by the check matrix. To avoid confusion, we define the $n \times 2n$ generator matrix $G_H$ of the classical code as

$$
G_H = H = \begin{bmatrix} \alpha^{(1)} & \beta^{(1)} \\ \vdots & \vdots \\ \alpha^{(n)} & \beta^{(n)} \end{bmatrix}. \tag{47}
$$

The code subspace $\mathcal{C} \subset \mathbb{F}_p^{2n}$ is

$$
\mathcal{C} = \left\{ c \in \mathbb{F}_p^{2n} \mid c = x\, G_H, \ x \in \mathbb{F}_p^n \right\}, \tag{48}
$$

where $x \in \mathbb{F}_p^n$ is an $n$-dimensional row vector. This classical code is a $[2n, n]_p$ code since the check matrix $H$ has rank $n$. We introduce the off-diagonal Lorentzian metric $\eta$ to the classical code $\mathcal{C}$:

$$
\eta = \begin{bmatrix} 0 & I_n \\ I_n & 0 \end{bmatrix}, \tag{49}
$$

where $I_n$ is the $n \times n$ identity. This metric is different from the symplectic form $W$ introduced earlier for $p \neq 2$ by (22). We denote the inner products with respect to the off-diagonal Lorentzian metric $\eta$ by $\odot$. Note that the norm of a codeword $c = (\alpha, \beta) \in \mathcal{C}$ with respect to the metric $\eta$ is always even:

$$
c \odot c \equiv c\, \eta\, c^T = 2\,\alpha \cdot \beta \in 2\mathbb{Z}, \tag{50}
$$

where the dot denotes the Euclidean inner product.

We define the dual code $\mathcal{C}^\perp$ with respect to the metric $\eta$ by

$$
\mathcal{C}^\perp = \left\{ c' \in \mathbb{F}_p^{2n} \mid c' \odot c = 0 \ \mathrm{mod}\, p, \ c \in \mathcal{C} \right\}. \tag{51}
$$

The classical code $\mathcal{C}$ is called self-orthogonal if $\mathcal{C} \subseteq \mathcal{C}^\perp$, and self-dual if $\mathcal{C} = \mathcal{C}^\perp$. Note that the notion of self-orthogonality and self-duality depends on the metric. In this section, we focus on the off-diagonal Lorentzian metric $\eta$.

For a $[2n, k']_p$ code $\mathcal{C}$ with the generator matrix $G_H$, one can take as the generator matrix $G_H^\perp$ of the dual code $\mathcal{C}^\perp$ any matrix such that

$$
G_H^\perp\, \eta\, G_H^T = 0 \qquad \mathrm{mod}\ p, \tag{52}
$$

and $\mathrm{rank}(G_H^\perp) = 2n - k'$. In the case of a self-orthogonal code $\mathcal{C}$, the following relation holds:

$$
G_H\, \eta\, G_H^T = 0 \qquad \mathrm{mod}\ p. \tag{53}
$$

If $k' = n$, the self-orthogonality condition (53) ensures self-duality $\mathcal{C} = \mathcal{C}^\perp$ as follows from the proposition below.

**Proposition 3.1**

Suppose that a $[[n,0]]_p$ qudit stabilizer code has a $n \times 2n$ check matrix $\mathsf{H}$. Then, the classical code with the generator matrix $G_\mathsf{H} = \mathsf{H}$ is self-dual with respect to the metric $\eta$ if and only if the check matrix satisfies the self-orthogonal condition: $\mathsf{H}\,\eta\,\mathsf{H}^T = 0 \bmod p$.

*Proof.* The generator matrix $G_\mathsf{H} = \mathsf{H}$ has rank $n$ due to the independence of the stabilizer generators. If the self-orthogonality condition (53) holds, the matrix $G_\mathsf{H}$ is also the generator matrix of the dual code $\mathcal{C}^\perp$ from (52) since it satisfies $\mathrm{rank}(G_\mathsf{H}) = n$. Then both the original code $\mathcal{C}$ and its dual $\mathcal{C}^\perp$ are generated by the matrix $G_\mathsf{H}$. This implies the classical code $\mathcal{C}$ is self-dual with respect to the metric $\eta$: $\mathcal{C} = \mathcal{C}^\perp$. On the other hand, if a classical code is self-dual, then the self-orthogonal condition is automatically satisfied. $\qquad\square$

The constructions of a lattice from a classical code are useful to search dense sphere packings and are well-studied by mathematicians (refer to [42] and the references therein). The simplest construction of them is called Construction A. The Construction A lattice $\Lambda(\mathcal{C})$ from a classical code $\mathcal{C}$ is defined by

$$\Lambda(\mathcal{C}) = \left\{ v/\sqrt{p} \,\middle|\, v \in \mathbb{Z}^{2n},\ v = c \pmod p,\ c \in \mathcal{C} \right\}. \tag{54}$$

The lattice $\Lambda(\mathcal{C})$ is a Lorentzian lattice with respect to the off-diagonal Lorentzian metric $\eta$ in (49). We use $\odot$ for the notation of the inner products between lattice vectors with the off-diagonal Lorentzian metric $\eta$ as in the case of a classical code $\mathcal{C}$.

By analogy with classical codes, we define the dual lattice with respect to the metric $\eta$ as follows:

$$\Lambda^* = \left\{ \lambda' \in \mathbb{R}^{n,n} \,\middle|\, \lambda' \odot \lambda \in \mathbb{Z},\ \lambda \in \Lambda \right\}. \tag{55}$$

The lattice $\Lambda$ is integral if and only if $\Lambda \subseteq \Lambda^*$ and self-dual if and only if $\Lambda = \Lambda^*$. We call the lattice $\Lambda$ even if and only if $\lambda \odot \lambda \in 2\mathbb{Z}$ for $\lambda \in \Lambda$.

The lattice $\Lambda(\mathcal{C})$ reduces to the classical code $\mathcal{C}$ by identifying $\lambda \sim \lambda + \sqrt{p}\,\mathbb{Z}^{2n}$, where $\lambda \in \Lambda(\mathcal{C})$. This implies that different codes give different lattices via Construction A. Then $\Lambda(\mathcal{C}) = \Lambda(\mathcal{C}')$ if and only if $\mathcal{C} = \mathcal{C}'$.

## 3.2 Even self-dual lattices

The above prescription defines the map between the classical code $\mathcal{C}$ derived from a qudit stabilizer code and the Lorentzian lattice $\Lambda(\mathcal{C})$, which associates the properties of the codes with those of the lattices. In this section, we describe the conditions for a classical code to give an even self-dual lattice via Construction A, some of which were obtained in [11]. For completeness we provide proofs in our notations. Then we translate the conditions into those on qudit stabilizer codes.

Starting with a qudit stabilizer code, we obtain a check matrix. We regard it as the generator matrix of a classical code $\mathcal{C}$ over $\mathbb{F}_p$ and construct a Lorentzian lattice $\Lambda(\mathcal{C})$. This construction connects a self-dual code $\mathcal{C}$ with the metric $\eta$ to a self-dual lattice $\Lambda(\mathcal{C})$ with the metric $\eta$. It can be summarized by the following proposition.

**Proposition 3.2** ( [11, Proposition 3.2])

For a prime $p$, the Construction A lattice $\Lambda(\mathcal{C})$ is self-dual with the off-diagonal Lorentzian metric $\eta$ if and only if a classical code $\mathcal{C}$ is self-dual with the Lorentzian metric $\eta$.

*Proof.* We first prove $\Lambda(\mathcal{C}^\perp) \supset \Lambda(\mathcal{C})^*$. Let us consider a vector $\lambda' = (\lambda_1', \lambda_2') \in \Lambda(\mathcal{C})^*$. A lattice vector in the Construction A lattice is given by $\lambda = (\lambda_1, \lambda_2) \in \Lambda(\mathcal{C})$ where

$$\lambda_1 = \frac{\alpha + p\,k_1}{\sqrt{p}}, \qquad \lambda_2 = \frac{\beta + p\,k_2}{\sqrt{p}}, \qquad k_1, k_2 \in \mathbb{Z}^n, \tag{56}$$

which is labeled by a codeword $c = (\alpha, \beta) \in \mathcal{C}$. Since the vector $\lambda'$ is in the dual lattice $\Lambda(\mathcal{C})^*$, the inner product with $\lambda \in \Lambda(\mathcal{C})$ must be an integer. Let $\lambda \in \Lambda(\mathcal{C})$ be $\lambda_1 = \sqrt{p}\, k_1$ and $\lambda_2 = \sqrt{p}\, k_2$. Then the inner product becomes

$$\lambda \odot \lambda' = \sqrt{p}\, \lambda_2' \cdot k_1 + \sqrt{p}\, \lambda_1' \cdot k_2\,, \qquad k_1, k_2 \in \mathbb{Z}^n\,. \tag{57}$$

To satisfy $\lambda \odot \lambda' \in \mathbb{Z}$, the lattice vector in the dual lattice has to be $\lambda' \in (\mathbb{Z}/\sqrt{p})^n$. Then the lattice vector $\lambda' = (\lambda_1', \lambda_2') \in \Lambda(\mathcal{C})^*$ can be written as the form

$$\lambda_1' = \frac{\alpha' + p\, k_1'}{\sqrt{p}}\,, \qquad \lambda_2' = \frac{\beta' + p\, k_2'}{\sqrt{p}}\,, \qquad k_1', k_2' \in \mathbb{Z}^n\,, \tag{58}$$

where $c' = (\alpha', \beta') \in \mathbb{F}_p^n \times \mathbb{F}_p^n$. The inner product between $\lambda \in \Lambda(\mathcal{C})$ and $\lambda' \in \Lambda(\mathcal{C})^*$ is

$$\lambda \odot \lambda' = \frac{\alpha' \cdot \beta + \alpha \cdot \beta'}{p} + (\alpha' \cdot k_2 + k_1' \cdot \beta + p\, k_1' \cdot k_2 + \alpha \cdot k_2' + k_1 \cdot \beta' + p\, k_1 \cdot k_2')\,. \tag{59}$$

The assumption $\lambda \odot \lambda' \in \mathbb{Z}$ gives us $c \odot c' = \alpha \cdot \beta' + \alpha' \cdot \beta = 0 \bmod p$. This implies $c' \in \mathcal{C}^\perp$ and $\lambda' \in \Lambda(\mathcal{C}^\perp)$.

To prove $\Lambda(\mathcal{C}^\perp) \subset \Lambda(\mathcal{C})^*$, we assume $\lambda \in \Lambda(\mathcal{C})$ and $\lambda' \in \Lambda(\mathcal{C}^\perp)$ take the same forms as (56) and (58), respectively, where $c = (\alpha, \beta) \in \mathcal{C}$ and $c' = (\alpha', \beta') \in \mathcal{C}^\perp$. Then, the inner product $\lambda \odot \lambda'$ given in (59) for any $\lambda \in \Lambda(\mathcal{C})$ becomes integer as $c \odot c' = \alpha \cdot \beta' + \alpha' \cdot \beta = 0 \bmod p$, which means $\lambda' \in \Lambda(\mathcal{C})^*$.

We have shown the lattice $\Lambda(\mathcal{C}^\perp)$ is the dual lattice of $\Lambda(\mathcal{C})$: $\Lambda^*(\mathcal{C}) = \Lambda(\mathcal{C}^\perp)$. Thus, for $\mathcal{C}$ a self-dual code $\mathcal{C} = \mathcal{C}^\perp$, the Construction A lattice is self-dual: $\Lambda^*(\mathcal{C}) = \Lambda(\mathcal{C})$. The inverse is also true because $\Lambda(\mathcal{C}) = \Lambda(\mathcal{C}')$ if and only if $\mathcal{C} = \mathcal{C}'$. Therefore, $\Lambda(\mathcal{C})$ is self-dual with respect to $\eta$ if and only if $\mathcal{C}$ is self-dual with respect to $\eta$. $\qquad\square$

Next, we construct an even lattice $\Lambda(\mathcal{C})$ with the Lorentzian metric from a classical code $\mathcal{C}$ with an appropriate property. This property is associated with the norm of a classical code $\mathcal{C}$ as in the following proposition. Note that there is a subtle difference between $p = 2$ and the other cases.

**Proposition 3.3** ( [11, Proposition 3.1])
For a prime $p \neq 2$, the Construction A lattice $\Lambda(\mathcal{C})$ is even with the Lorentzian metric $\eta$ if and only if a classical code $\mathcal{C}$ is self-orthogonal with the off-diagonal Lorentzian metric $\eta$.

*Proof.* Suppose that a codeword $c = (\alpha, \beta) \in \mathcal{C}$. The Construction A lattice is given by $\lambda = (\lambda_1, \lambda_2) \in \Lambda(\mathcal{C})$ where

$$\lambda_1 = \frac{\alpha + p\, k_1}{\sqrt{p}}\,, \qquad \lambda_2 = \frac{\beta + p\, k_2}{\sqrt{p}}\,, \qquad k_1, k_2 \in \mathbb{Z}^n\,. \tag{60}$$

The norm of the lattice vector is

$$\lambda \odot \lambda = \frac{2}{p}\left(\alpha \cdot \beta + p\, \alpha \cdot k_2 + p\, \beta \cdot k_1 + p^2\, k_1 \cdot k_2\right)\,. \tag{61}$$

Let $\mathcal{C}$ be a self-orthogonal code. Then the codeword satisfies $c \odot c = 2\, \alpha \cdot \beta = 0 \bmod p$. This implies $\alpha \cdot \beta \in p\,\mathbb{Z}$ since $\alpha \cdot \beta \in \mathbb{Z}$ and $p$ and $2$ are coprime for an odd prime $p \neq 2$. Thus, we conclude the norm of the lattice vector is even. On the other hand, let $\Lambda(\mathcal{C})$ be even with respect to the metric $\eta$. Then we obtain $(\alpha \cdot \beta)/p \in \mathbb{Z}$. This implies $c \odot c = 2\, \alpha \cdot \beta = 0 \bmod p$. The relation $(c + c') \odot (c + c') = c \odot c + c' \odot c' + 2\, c \odot c'$ for $c, c' \in \mathcal{C}$ ensures self-orthogonality of the classical code $\mathcal{C}$ for an odd prime $p \neq 2$: $c \odot c' \in p\,\mathbb{Z}$ for any pair of $c, c' \in \mathcal{C}$. $\qquad\square$

**Proposition 3.4**
For $p = 2$, the Construction A lattice $\Lambda(\mathcal{C})$ is even with respect to the off-diagonal Lorentzian metric $\eta$ if and only if a classical code $\mathcal{C}$ is doubly-even with respect to the metric $\eta$: $c \odot c = 0$ mod 4 where $c \in \mathcal{C}$.

*Proof.* Suppose that a lattice vector $\lambda = (\lambda_1, \lambda_2)$ in the Construction A lattice $\Lambda(\mathcal{C})$ is

$$\lambda_1 = \frac{\alpha + 2 k_1}{\sqrt{2}}, \qquad \lambda_2 = \frac{\beta + 2 k_2}{\sqrt{2}}, \qquad k_1, k_2 \in \mathbb{Z}^n, \tag{62}$$

where $c = (\alpha, \beta) \in \mathcal{C}$ is a codeword. The norm of this vector is

$$\lambda \odot \lambda = \alpha \cdot \beta + 2 \alpha \cdot k_2 + 2 \beta \cdot k_1 + 4 k_1 \cdot k_2. \tag{63}$$

Let $\mathcal{C}$ be doubly-even: $c \odot c = 2 \alpha \cdot \beta = 0$ mod 4. Then we have $\alpha \cdot \beta = 0$ mod 2, so the norm of a lattice vector is even. On the other hand, suppose that the Construction A lattice is even. Then, it results in $\alpha \cdot \beta = 0$ mod 2, which is equivalent to doubly-evenness: $c \odot c = 2 \alpha \cdot \beta = 0$ mod 4. $\square$

Proposition 3.2 and Proposition 3.3 or 3.4 lead to the following theorem that ensures that a class of qudit stabilizer codes yields Lorentzian even self-dual lattices via Construction A.

**Theorem 3.5** ( [11, Proposition 3.3] for $p \neq 2$)
For a prime $p \neq 2$, a self-dual code $\mathcal{C}$ with the off-diagonal Lorentzian metric $\eta$ gives an even self-dual lattice $\Lambda(\mathcal{C})$ with the metric $\eta$ via Construction A. For $p = 2$, a doubly-even self-dual code $\mathcal{C}$ with the metric $\eta$ endows an even self-dual lattice $\Lambda(\mathcal{C})$ with the metric $\eta$.

We now combine the above theorem and Proposition 3.1 to obtain the conditions for a qudit stabilizer code to give a Lorentzian even self-dual lattice.

**Corollary 3.6**
Suppose that a $[[n, 0]]_p$ qudit stabilizer code has an $n \times 2n$ check matrix $\mathsf{H}$ satisfying $\mathsf{H} \eta \mathsf{H}^T = 0$ mod $p$. For an odd prime $p \neq 2$, a $p$-ary classical code $\mathcal{C}$ generated by the matrix $G_{\mathsf{H}} = \mathsf{H}$ prepares an even self-dual lattice $\Lambda(\mathcal{C})$ with yields to the off-diagonal Lorentzian metric $\eta$.

For $p = 2$, we must consider the additional condition to ensure doubly-evenness of the classical code $\mathcal{C}$. It also reduces to a simple condition for the generator matrix.

**Corollary 3.7**
Suppose that a $[[n, 0]]_2$ binary stabilizer code has an $n \times 2n$ check matrix $\mathsf{H}$ that satisfies $\mathsf{H} \eta \mathsf{H}^T = 0$ mod 2 and $\text{diag}(\mathsf{H} \eta \mathsf{H}^T) = 0$ mod 4. Then, a binary classical code $\mathcal{C}$ generated by the matrix $G_{\mathsf{H}} = \mathsf{H}$ gives an even self-dual lattice $\Lambda(\mathcal{C})$ with respect to the metric $\eta$.

*Proof.* We already know that the condition $\mathsf{H} \eta \mathsf{H}^T = 0$ mod 2 guarantees that the classical code generated by $G_{\mathsf{H}} = \mathsf{H}$ is self-dual from Proposition 3.1. Thus, we only have to verify that the assumptions $\mathsf{H} \eta \mathsf{H}^T = 0$ mod 2 and $\text{diag}(\mathsf{H} \eta \mathsf{H}^T) = 0$ mod 4 ensure the classical code to be doubly-even. Let $c^{(i)}$ $(i = 1, 2, \cdots, n)$ be the $i$-th row of the generator matrix $G_{\mathsf{H}}$. These vectors form a basis of the code subspace $\mathcal{C}$. Then, a codeword $c$ is written as $c = \sum_i s_i c^{(i)}$ and its norm is given by

$$c \odot c = \sum_{i,j} s_i s_j c^{(i)} \odot c^{(j)} = \sum_i s_i^2 c^{(i)} \odot c^{(i)} + 2 \sum_{i < j} s_i s_j c^{(i)} \odot c^{(j)}. \tag{64}$$

The condition $\mathsf{H} \eta \mathsf{H}^T = 0$ mod 2 reduces to $c^{(i)} \odot c^{(j)} = 0$ mod 2. The other condition $\text{diag}(\mathsf{H} \eta \mathsf{H}^T) = 0$ mod 4 implies $c^{(i)} \odot c^{(i)} = 0$ mod 4. From (64), the norm $c \odot c$ becomes a multiple of 4: $c \odot c \in 4\mathbb{Z}$ for any codeword $c \in \mathcal{C}$. That is, the classical code $\mathcal{C}$ is doubly-even. $\square$

While we have discussed the construction of Lorentzian lattice for qubit cases and qudit cases in parallel, we emphasize the difference between them mentioned in section 2.2.3. The Clifford group preserves the group structure of the Pauli group, so a stabilizer group is kept abelian under the action of the Clifford group. In the language of a check matrix, this property implies that the symplectic form $\mathsf{W}$ is invariant with the Clifford group transformation. For example, let us consider the Hadamard transformation: $X_p \to Z_p$, $Z_p \to X_p^{-1}$. If the Hadamard transformation acts on the $i$-th qudit, the $i$-th column and the $(i + n)$-th column in the check matrix are swapped with $-1$ on one side. It keeps the symplectic form invariant. However, the Hadamard transformation does change the inner products with respect to the off-diagonal metric $\eta$. Therefore, the Clifford group does not preserve the structure of the Lorentzian metric $\eta$ for an odd prime $p$. On the other hand, for qubits ($p = 2$), the symplectic form $\mathsf{W}$ coincides with the metric $\eta$ introduced later, so the Clifford group also preserves the metric $\eta$ in this case [4].

## 3.3 CSS construction

We have described the conditions for a qudit stabilizer code to give a Lorentzian even self-dual lattice. We now explain that the CSS codes reviewed in section 2.3 satisfy the conditions and discuss an explicit example of the construction of lattices from CSS codes, which we will use heavily later in this paper.

We start with a classical $[n, k]_p$ code $C$ with a generator matrix $G_C$ and a parity check matrix $H_C$. Then, the dual code $C^\perp$ has the generator matrix $H_C$ and the parity check matrix $G_C$. Note that we do not require the code $C$ to be self-orthogonal or self-dual. As a special case of $C_X$ and $C_Z$ satisfying (36), we choose $C_X = C$ and $C_Z = C^\perp$. Then, the code $C_X = C$ has the generator matrix $G_X = G_C$ and the parity check matrix $H_X = H_C$. On the other hand, the code $C_Z = C^\perp$ has the generator matrix $G_Z = H_C$ and the parity check matrix $H_Z = G_C$. For this choice, the condition (37) reduces to $G_C H_C^T = 0 \mod p$, and it is satisfied due to the relation (30) between the generator matrix and the parity check matrix. Then, the $n \times 2n$ check matrix of the CSS code is as follows:

$$\mathsf{H}_{(C,C^\perp)} = \begin{bmatrix} H_C & 0 \\ 0 & G_C \end{bmatrix}. \tag{65}$$

We denote a classical code generated by the matrix $G_{\mathsf{H}} = \mathsf{H}_{(C,C^\perp)}$ as $\mathcal{C}$:

$$\mathcal{C} = \left\{ (c_1, c_2) \in \mathbb{F}_p^n \times \mathbb{F}_p^n \mid c_1 \in C^\perp, \, c_2 \in C \right\}. \tag{66}$$

The following theorem verifies that the CSS code $\mathcal{C}$ leads to an even self-dual lattice through Construction A, giving explicit examples of the construction of a Lorentzian even self-dual lattice from a qudit stabilizer code.

**Theorem 3.8**

Suppose that a CSS code has a check matrix (65) with a classical $[n, k]_p$ code $C$ and the dual code $C^\perp$. Let $\mathcal{C}$ be the classical code with the generator matrix $\mathsf{H}_{(C,C^\perp)}$. Then, the Construction A lattice $\Lambda(\mathcal{C})$ is even self-dual with respect to the metric $\eta$.

*Proof.* For a prime $p \neq 2$, all we have to do is to check self-duality of the code $\mathcal{C}$ with respect to the Lorentzian metric $\eta$. The dual code with the metric $\eta$ is defined by

$$\mathcal{C}^\perp = \left\{ (c_1', c_2') \in \mathbb{F}_p^n \times \mathbb{F}_p^n \mid (c_1', c_2') \odot (c_1, c_2) = 0 \mod p, \, (c_1, c_2) \in \mathcal{C} \right\}. \tag{67}$$

Since the metric is given by (49), this implies that $(c_1', c_2')$ is in the dual code if and only if $c_1' \cdot c_2 + c_2' \cdot c_1 = 0 \mod p$ for any $c_1 \in C^\perp$ and $c_2 \in C$. Thus, the above definition reduces to the

following condition: $c_1' \cdot c_2 = 0 \bmod p$ and $c_2' \cdot c_1 = 0 \bmod p$. Equivalently, $c_1' G_C^T = c_2' H_C^T = 0 \bmod p$. This means $c_1' \in C^\perp$ and $c_2' \in C$ through (33):

$$\mathcal{C}^\perp = \left\{ (c_1', c_2') \in \mathbb{F}_p^n \times \mathbb{F}_p^n \,|\, c_1' \in C^\perp, c_2' \in C \right\} \equiv \mathcal{C}. \tag{68}$$

Therefore, a classical code obtained through the CSS construction is self-dual with respect to the metric $\eta$. For a prime $p \neq 2$, Theorem 3.5 states the CSS code $\mathcal{C}$ generated by a classical $p$-ary self-dual code gives an even self-dual lattice $\Lambda(\mathcal{C})$.

To ensure that the Construction A lattice is even for $p = 2$, an additional condition should be imposed. In this case, we require the CSS code $\mathcal{C}$ to be doubly-even with respect to the metric $\eta$ as dictated by Theorem 3.5. Then, for a classical binary $[n, k]_2$ code $C$, we have

$$c \odot c = 2c_1 \cdot c_2 \in 4\mathbb{Z}, \qquad c_1 \in C^\perp, \; c_2 \in C, \tag{69}$$

where $c = (c_1, c_2) \in \mathcal{C}$ and the dot denotes the Euclidean inner product on the classical code $C$. The condition for a doubly-even code is $c_1 \cdot c_2 = 0 \bmod 2$ and this is satisfied as an inner product between the code $C$ and the dual code $C^\perp$ vanishes modulo 2. There are no additional requirements for doubly-evenness in the case of the CSS construction.

Therefore, the classical code $\mathcal{C}$ starting with a classical $[n, k]_p$ code $C$ becomes self-dual for a prime $p$ and doubly-even for $p = 2$. Hence, the Construction A lattice from the CSS code is even and self-dual with respect to the off-diagonal Lorentzian metric $\eta$. $\qquad \square$

We can choose a classical code $C$ to be self-dual. Following the above prescription, we give the CSS code constructed from a pair of classical codes $C_X$, $C_Z$ such that $C_X = C_Z = C$. Then, the check matrix of the CSS code is

$$\mathsf{H}_{(C,C)} = \begin{bmatrix} H_C & 0 \\ 0 & H_C \end{bmatrix}. \tag{70}$$

The classical code with the generator matrix $G_\mathsf{H} = \mathsf{H}_{(C,C)}$ is given by

$$\mathcal{C} = \left\{ (c_1, c_2) \in \mathbb{F}_p^n \times \mathbb{F}_p^n \,|\, c_1, c_2 \in C \right\}. \tag{71}$$

This is an example of the construction dictated in Theorem 3.8. In this case, of course, the classical code $\mathcal{C}$ gives an even self-dual lattice via Construction A. In section 5, we will consider the averaged theory over the CSS codes defined from a classical self-dual code.

**Corollary 3.9**
Suppose that a CSS code has a check matrix (70) with a classical $[n, n/2]_p$ self-dual code $C$. Let $\mathcal{C}$ be the classical code with the generator matrix $\mathsf{H}_{(C,C)}$. Then, the Construction A lattice $\Lambda(\mathcal{C})$ is even self-dual with respect to the off-diagonal Lorentzian metric $\eta$.

# 4 Narain code CFTs

We have seen that an even self-dual Lorentzian lattice can be constructed from a qudit stabilizer code with appropriate conditions via Construction A. In this section, we assume that $\Lambda(\mathcal{C})$ is a Lorentzian even self-dual lattice obtained through Construction A. We associate the lattice with a Narain CFT [48, 49], a free boson theory with a torus target space. We refer to the Narain CFTs constructed from codes as Narain code CFTs.

## 4.1 Construction of Narain CFTs

From a qudit stabilizer code, we can construct a Narain lattice, i.e., an even self-dual lattice as in Corollary 3.6 and 3.7. Naively, a Narain CFT is given by choosing the Construction A lattice as the momentum lattice. However, there is a subtlety in this naive construction. The Construction A lattices are equipped with an off-diagonal Lorentzian metric $\eta$, so they are given in the coordinates:

$$\lambda = (\lambda_1, \lambda_2) = \left( \frac{p_L + p_R}{\sqrt{2}}, \frac{p_L - p_R}{\sqrt{2}} \right) \in \Lambda(\mathcal{C}), \tag{72}$$

rather than the coordinates of the left- and right-moving momentum $(p_L, p_R)$. The norm of $\lambda = (\lambda_1, \lambda_2) \in \Lambda(\mathcal{C})$ with respect to the off-diagonal Lorentzian metric $\eta$ is

$$(\lambda_1, \lambda_2) \odot (\lambda_1, \lambda_2) = p_L^2 - p_R^2 = (p_L, p_R) \circ (p_L, p_R), \tag{73}$$

where we follow the notation of Polchinski's textbook [50,51]. This is associated with a natural metric for the left- and right-moving momentum in the Narain lattices:

$$\widetilde{\eta} = \begin{bmatrix} I_n & 0 \\ 0 & -I_n \end{bmatrix}, \tag{74}$$

where $I_n$ is the $n \times n$ identity matrix. To show it explicitly, we have to move onto the momentum basis by the orthogonal transformation:

$$(p_L, p_R) = (\lambda_1, \lambda_2) P, \qquad P = \frac{1}{\sqrt{2}} \begin{bmatrix} I_n & I_n \\ I_n & -I_n \end{bmatrix}. \tag{75}$$

The left- and right-moving momentum are given by points $(p_L, p_R) \in \widetilde{\Lambda}(\mathcal{C})$ in the momentum lattice $\widetilde{\Lambda}(\mathcal{C})$. The vertex operators in the Narain code CFTs are given by

$$V_{p_L, p_R}(z, \bar{z}) =: e^{i p_L X_L(z) + i p_R X_R(\bar{z})} :, \tag{76}$$

where $(p_L, p_R) \in \widetilde{\Lambda}(\mathcal{C})$. We omit the cocycle factors, which do not matter for our analysis. These operators correspond to the momentum states $|p_L, p_R\rangle$ via the state-operator isomorphism. We have the oscillators $\alpha_k^i$ and $\tilde{\alpha}_k^i$ ($i = 1, 2, \cdots, n$) that satisfy the following algebra:

$$[\alpha_k^i, \alpha_l^j] = [\tilde{\alpha}_k^i, \tilde{\alpha}_l^j] = k \, \delta_{k+l,0} \, \delta^{i,j}, \qquad k, l \in \mathbb{Z}. \tag{77}$$

The Hilbert space of the Narain code CFT is given by

$$\mathcal{H}(\mathcal{C}) = \left\{ \alpha_{-k_1}^{i_1} \cdots \alpha_{-k_r}^{i_r} \, \tilde{\alpha}_{-l_1}^{j_1} \cdots \tilde{\alpha}_{-l_s}^{j_s} \, |p_L, p_R\rangle \mid (p_L, p_R) \in \widetilde{\Lambda}(\mathcal{C}) \right\}, \tag{78}$$

with $k_1, \cdots, k_r \in \mathbb{Z}_{>0}$ and $l_1, \cdots, l_s \in \mathbb{Z}_{>0}$. Therefore, we arrive at the following proposition:

**Proposition 4.1**
Let $\Lambda(\mathcal{C})$ be the Construction A lattice that is even self-dual with respect to the off-diagonal Lorentzian metric $\eta$. Suppose that $\widetilde{\Lambda}(\mathcal{C})$ is the lattice obtained by the orthogonal transformation (75) of the Construction A lattice $\Lambda(\mathcal{C})$. Then, a Narain CFT is provided by giving the left- and right-moving momenta as $(p_L, p_R) \in \widetilde{\Lambda}(\mathcal{C})$.

By combining this proposition with Corollary 3.6 and 3.7, we finally get the following theorems that summarize our construction of the Narain code CFTs:

**Theorem 4.2**
Suppose that a $[[n,0]]_p$ qudit stabilizer code has an $n \times 2n$ check matrix $\mathsf{H}$ satisfying $\mathsf{H}\,\eta\,\mathsf{H}^T = 0$ mod $p$. Let $\mathcal{C}$ be a classical code generated by the matrix $G_\mathsf{H} = \mathsf{H}$. For an odd prime $p \neq 2$, the Construction A lattice $\widetilde{\Lambda}(\mathcal{C})$ followed by the orthogonal transformation (75) provides a Narain CFT by giving the left- and right-moving momenta as $(p_L, p_R) \in \widetilde{\Lambda}(\mathcal{C})$.

**Theorem 4.3**
Suppose that a $[[n,0]]_2$ qubit stabilizer code has an $n \times 2n$ check matrix $\mathsf{H}$ that satisfies $\mathsf{H}\,\eta\,\mathsf{H}^T = 0$ mod $2$ and $\mathrm{diag}(\mathsf{H}\,\eta\,\mathsf{H}^T) = 0$ mod $4$. Let $\mathcal{C}$ be a binary classical code generated by the matrix $G_\mathsf{H} = \mathsf{H}$. Then, the Construction A lattice $\widetilde{\Lambda}(\mathcal{C})$ followed by the orthogonal transformation (75) provides a Narain CFT by giving the left- and right-moving momenta as $(p_L, p_R) \in \widetilde{\Lambda}(\mathcal{C})$.

The torus partition function of the resulting Narain code CFT is as follows:

$$Z_{\mathcal{C}}(\tau, \bar{\tau}) = \mathrm{Tr}_{\mathcal{H}(\mathcal{C})}\, q^{L_0 - \frac{n}{24}}\, \bar{q}^{\bar{L}_0 - \frac{n}{24}} = \frac{\Theta_{\widetilde{\Lambda}(\mathcal{C})}(\tau, \bar{\tau})}{|\eta(\tau)|^{2n}}, \tag{79}$$

where $\eta(\tau)$ is the Dedekind eta function. The lattice theta function of the Narain lattice is

$$\Theta_{\widetilde{\Lambda}(\mathcal{C})}(\tau, \bar{\tau}) = \sum_{p \in \widetilde{\Lambda}(\mathcal{C})} q^{\frac{p_L^2}{2}}\, \bar{q}^{\frac{p_R^2}{2}}, \tag{80}$$

where $q = e^{2\pi i \tau}$ and $\tau = \tau_1 + i\tau_2$ is the modulus of the torus. Note that the CFT partition function (79) explicitly depends on the decomposition of the lattice $\widetilde{\Lambda}(\mathcal{C})$ into the left-moving momentum $p_L$ and the right-moving momentum $p_R$. However, the inner product does not depend on the coordinate (73), so $\widetilde{\Lambda}(\mathcal{C})$ is also even and self-dual with respect to the diagonal Lorentzian metric $\widetilde{\eta}$. Therefore, the modular invariance of the partition function constructed from the momentum lattice $\widetilde{\Lambda}(\mathcal{C})$ follows directly from that the Construction A lattice $\Lambda(\mathcal{C})$ is even self-dual.

## 4.2 Partition function

We have obtained the direct connection (79) between the partition function and the lattice theta function. Both of these quantities characterize each spectrum. There is also a quantity that measures the spectrum of codes, which is called the enumerator polynomial. The construction above gives a simple relation between the spectrum of Narain CFTs, lattices, and codes. Using this relationship, it is straightforward to calculate the partition function of the Narain CFT in terms of the code enumerator polynomial. In what follows, we will determine the partition function of the Narain CFT constructed from a qudit code and explain how each spectrum is tied together.

The Construction A lattice has a concrete representation by a codeword $c = (\alpha, \beta) \in \mathcal{C}$ where $\alpha = (\alpha_1, \cdots, \alpha_n) \in \mathbb{F}_p^n$ and $\beta = (\beta_1, \cdots, \beta_n) \in \mathbb{F}_p^n$:

$$\lambda_1 = \frac{\alpha + p\,k_1}{\sqrt{p}}, \qquad \lambda_2 = \frac{\beta + p\,k_2}{\sqrt{p}}, \qquad k_1, k_2 \in \mathbb{Z}^n. \tag{81}$$

Therefore, the partition function of the Narain code CFT can be expressed in terms of codewords $c = (\alpha, \beta) \in \mathcal{C}$:

$$Z_{\mathcal{C}}(\tau, \bar{\tau}) = \frac{1}{|\eta(\tau)|^{2n}} \sum_{(\alpha, \beta) \in \mathcal{C}} \sum_{k_1, k_2 \in \mathbb{Z}^n} q^{\frac{p}{4}\left(\frac{\alpha+\beta}{p} + k_1 + k_2\right)^2} \bar{q}^{\frac{p}{4}\left(\frac{\alpha-\beta}{p} + k_1 - k_2\right)^2}. \tag{82}$$

We can associate the partition function with the complete enumerator polynomial of the code $\mathcal{C}$. The complete enumerator polynomial of a code $\mathcal{C}$ is defined by ( [52,53])

$$W_{\mathcal{C}}(\{x_{ab}\}) = \sum_{c \in \mathcal{C}} \prod_{(a,b) \in \mathbb{F}_p \times \mathbb{F}_p} x_{ab}^{\mathrm{wt}_{ab}(c)}, \tag{83}$$

where $\mathrm{wt}_{ab}(c)$ is the number of components $c_i = (\alpha_i, \beta_i) \in \mathbb{F}_p \times \mathbb{F}_p$ that equal to $(a, b) \in \mathbb{F}_p \times \mathbb{F}_p$ for a codeword $c \in \mathcal{C}$:

$$\mathrm{wt}_{ab}(c) = |\{i \mid c_i = (a, b)\}|, \tag{84}$$

which is called the *composition* of $c \in \mathcal{C}$ in [40,53]. The complete enumerator polynomial of the dual code $\mathcal{C}^{\perp}$ is uniquely determined by the one of $\mathcal{C}$. We obtain the complete enumerator polynomial of the dual code $\mathcal{C}^{\perp}$ from the MacWilliams identity [54,55] (see also Theorem 10 of Chapter 5 in [40] and Example 2.2.7 in [53]):

$$W_{\mathcal{C}^{\perp}}(\{x_{ab}\}) = W_{\mathcal{C}}(\{\tilde{x}_{ab}\}), \tag{85}$$

where for $v = (a, b) \in \mathbb{F}_p \times \mathbb{F}_p$

$$\tilde{x}_v = \frac{1}{p} \sum_{w \in \mathbb{F}_p \times \mathbb{F}_p} e^{\frac{2\pi i}{p} w \eta_2 v^T} x_w, \tag{86}$$

with the non-degenerate symmetric bilinear form $\eta_2$ on $\mathbb{F}_p \times \mathbb{F}_p$:

$$\eta_2 = \begin{bmatrix} 0 & 1 \\ 1 & 0 \end{bmatrix}. \tag{87}$$

We can also write the relation as

$$x_v = \frac{1}{p} \sum_{w \in \mathbb{F}_p \times \mathbb{F}_p} e^{-\frac{2\pi i}{p} w \eta_2 v^T} \tilde{x}_w. \tag{88}$$

Then, for a self-dual code $\mathcal{C} = \mathcal{C}^{\perp}$, the complete enumerator polynomial is invariant under the change of variables $x_{a_i b_i} \longleftrightarrow \tilde{x}_{a_i b_i}$. The invariance of the complete enumerator polynomial is closely related to the modular invariance for the partition functions of Narain code CFTs. We will see it later in this section.

We can explicitly relate the complete enumerator polynomial to the partition function.

**Proposition 4.4**
Let $\mathcal{C} \subset \mathbb{F}_p^n \times \mathbb{F}_p^n$ be a classical code whose complete enumerator polynomial $W_{\mathcal{C}}$ is given by (83). Then, the partition function of the Narain CFT constructed from the code $\mathcal{C}$ is

$$Z_{\mathcal{C}}(\tau, \bar{\tau}) = \frac{\Theta_{\tilde{\Lambda}(\mathcal{C})}(\tau, \bar{\tau})}{|\eta(\tau)|^{2n}} = \frac{1}{|\eta(\tau)|^{2n}} W_{\mathcal{C}}(\{\psi_{ab}\}), \tag{89}$$

where the variables $x_{ab}$ in the complete enumerator polynomial are replaced by

$$\psi_{ab}(\tau, \bar{\tau}) = \sum_{k_1, k_2 \in \mathbb{Z}} q^{\frac{p}{4}\left(\frac{a+b}{p} + k_1 + k_2\right)^2} \bar{q}^{\frac{p}{4}\left(\frac{a-b}{p} + k_1 - k_2\right)^2}. \tag{90}$$

*Proof.* We start with the complete enumerator polynomial

$$W_{\mathcal{C}}(\{\psi_{ab}\}) = \sum_{c \in \mathcal{C}} \prod_{(a,b) \in \mathbb{F}_p \times \mathbb{F}_p} \psi_{ab}(\tau, \bar{\tau})^{\mathrm{wt}_{ab}(c)}. \tag{91}$$

The composition of a codeword $c \in \mathcal{C}$ is given by the sum of $\mathrm{wt}_{ab}(c_i)$ for each component:

$$\mathrm{wt}_{ab}(c) = \sum_{i=1}^{n} \mathrm{wt}_{ab}(c_i), \tag{92}$$

where, for each component of a codeword, we define

$$\mathrm{wt}_{ab}(c_i) = \begin{cases} 1, & c_i = (a, b), \\ 0, & c_i \neq (a, b). \end{cases} \tag{93}$$

Then, we have

$$\begin{aligned} W_{\mathcal{C}}(\{\psi_{ab}\}) &= \sum_{c \in \mathcal{C}} \prod_{i=1}^{n} \prod_{(a,b) \in \mathbb{F}_p \times \mathbb{F}_p} \psi_{ab}(\tau, \bar{\tau})^{\mathrm{wt}_{ab}(c_i)} \\ &= \sum_{(\alpha,\beta) \in \mathcal{C}} \prod_{i=1}^{n} \psi_{\alpha_i \beta_i}(\tau, \bar{\tau}) \\ &= \sum_{(\alpha,\beta) \in \mathcal{C}} \sum_{k_1, k_2 \in \mathbb{Z}^n} q^{\frac{p}{4}\left(\frac{\alpha+\beta}{p}+k_1+k_2\right)^2} \bar{q}^{\frac{p}{4}\left(\frac{\alpha-\beta}{p}+k_1-k_2\right)^2} \\ &= \Theta_{\tilde{\Lambda}(\mathcal{C})}(\tau, \bar{\tau}). \end{aligned} \tag{94}$$

The lattice theta function of the Construction A lattice from a classical code $\mathcal{C}$ appears. From (79), we divide the complete enumerator polynomial by $|\eta(\tau)|^{2n}$ to show the statement. □

It is useful to write the function $\psi_{ab}$ as

$$\psi_{ab}(\tau, \bar{\tau}) = \Theta_{a+b,p}(\tau) \bar{\Theta}_{a-b,p}(\bar{\tau}) + \Theta_{a+b-p,p}(\tau) \bar{\Theta}_{a-b-p,p}(\bar{\tau}), \tag{95}$$

where $(a, b) \in \mathbb{F}_p \times \mathbb{F}_p$ and $\Theta_{m,k}(\tau)$ is the theta function

$$\Theta_{m,k}(\tau) = \sum_{n \in \mathbb{Z}} q^{k\left(n+\frac{m}{2k}\right)^2}. \tag{96}$$

For an integer $m \in \mathbb{Z}$, the modular transformations of the theta functions are

$$\Theta_{m,k}(\tau+1) = e^{2\pi i \frac{m^2}{4k}} \Theta_{m,k}(\tau), \tag{97}$$

$$\Theta_{m,k}(-1/\tau) = \sqrt{-i\tau} \sum_{m' \in \mathbb{Z}_{2k}} M_{mm'}^{(k)} \Theta_{m',k}(\tau), \tag{98}$$

where $M_{mm'}^{(k)} = \frac{1}{\sqrt{2k}} e^{-2\pi i \frac{mm'}{2k}}$.

Let us return to the modular invariance for the partition functions of Narain code CFTs. We can derive it directly from the property of the code $\mathcal{C}$. To see it, let us focus on the modular property of the lattice theta function in (89) since the modular transformation of the Dedekind eta function is given by

$$\eta(\tau+1) = e^{2\pi i \frac{1}{24}} \eta(\tau), \qquad \eta(-1/\tau) = \sqrt{-i\tau}\, \eta(\tau). \tag{99}$$

It is straightforward to see that the function $\psi_{ab}$ behaves as follows under the modular transformation:

$$\psi_{ab}(\tau,\bar{\tau}) \to e^{2\pi i \frac{ab}{p}} \psi_{ab}(\tau,\bar{\tau}), \qquad\qquad (\tau \to \tau+1),$$

$$\psi_{ab}(\tau,\bar{\tau}) \to \frac{|-i\tau|}{p} \sum_{w_1,w_2 \in \mathbb{F}_p} e^{-\frac{2\pi i}{p}(w_1,w_2)\eta_2(a,b)^T} \psi_{w_1 w_2}(\tau,\bar{\tau}), \qquad (\tau \to -1/\tau). \tag{100}$$

Under the modular transformation $\tau \to \tau+1$, the lattice theta function behaves as

$$\Theta_{\widetilde{\Lambda}(\mathcal{C})}(\tau+1,\bar{\tau}+1) = W_{\mathcal{C}}(\{e^{2\pi i \frac{ab}{p}}\psi_{ab}\}) = \sum_{c \in \mathcal{C}} e^{\frac{2\pi i}{p}\sum_{a,b \in \mathbb{F}_p} ab\, \mathrm{wt}_{ab}(c)} \prod_{(a,b) \in \mathbb{F}_p \times \mathbb{F}_p} \psi_{ab}^{\mathrm{wt}_{ab}(c)}. \tag{101}$$

Our Narain code CFTs are based on doubly-even self-dual codes for $p=2$ and self-dual codes for odd prime $p$. Then, the norm $c \odot c = 2\alpha \cdot \beta = 2\sum_{a,b \in \mathbb{F}_p} ab\, \mathrm{wt}_{ab}(c)$ becomes a multiple of 4 for $p=2$ and a multiple of $p$ for odd prime $p$. Since 2 and $p$ are coprime for odd prime $p$, we have

$$\sum_{a,b \in \mathbb{F}_p} ab\, \mathrm{wt}_{ab}(c) = 0 \qquad \mathrm{mod}\ p. \tag{102}$$

Therefore, the lattice theta function is invariant under the modular transformation $\tau \to \tau+1$ from (101). Note that the invariance of the lattice theta function directly follows from doubly-evenness for $p=2$ and self-orthogonality for odd prime $p$. From the modular property (99) of the Dedekind eta function, we obtain the immediate consequence that the partition function is also invariant under $\tau \to \tau+1$.

On the other hand, the lattice theta function transforms as follows under the modular transformation $\tau \to -1/\tau$:

$$\Theta_{\widetilde{\Lambda}(\mathcal{C})}(-1/\tau,-1/\bar{\tau}) = |-i\tau|^n W_{\mathcal{C}}(\{\Psi_{ab}\}), \tag{103}$$

where

$$\Psi_{ab}(\tau,\bar{\tau}) = \frac{1}{p} \sum_{w_1,w_2 \in \mathbb{F}_p} e^{-\frac{2\pi i}{p}(w_1,w_2)\eta_2(a,b)^T} \psi_{w_1 w_2}(\tau,\bar{\tau}), \tag{104}$$

where we use the fact that the complete enumerator polynomial is a homogeneous polynomial of degree $n$. We observe that the relation between $\Psi_{ab}$ and $\psi_{w_1 w_2}$ is same as one between $x_v$ and $\tilde{x}_w$ in (88). Hence, the MacWilliams identity ensures that the complete enumerator polynomial is invariant under the linear transformation $\Psi_{ab} \leftrightarrow \psi_{ab}$ for a self-dual code $\mathcal{C}$: $W_{\mathcal{C}}(\{\Psi_{ab}\}) = W_{\mathcal{C}}(\{\psi_{ab}\})$. We conclude that, under the modular transformation $\tau \to -1/\tau$, the lattice theta function behaves as $W_{\mathcal{C}}(\{\psi_{ab}\}) \to |-i\tau|^n W_{\mathcal{C}}(\{\psi_{ab}\})$. The term $|-i\tau|^n$ that appears from the complete enumerator polynomial cancels with the one from the modular transformation (99) of the Dedekind eta function in the partition function. Therefore, the partition functions of our Narain code CFTs are invariant under $\tau \to -1/\tau$.

In this section, we have connected the properties and quantities of codes, lattices, and CFTs. For example, the complete enumerator polynomial determines the lattice theta function of the Construction A lattice and the partition function of the Narain code CFT. We show a list summarizing the main relations in table 1 while omitting some items for quantum codes because it does not matter in our construction.

Table 1: The properties of codes, lattices and CFTs.

| Quantum code | Classical code | Lattice | CFT |
|---|---|---|---|
| number of qudits | length | rank | central charge |
| stabilizer element $g(\alpha,\beta)$ | codeword $c$ | lattice vector $\lambda$ | momentum $(p_L, p_R)$ |
| | norm $c \odot c$ | length $\lambda \odot \lambda$ | spin $h - \bar{h}$ |
| | $W_C(\{x_{ab}\})$ | $\Theta_{\tilde{\Lambda}(C)}(\tau,\bar{\tau})$ | $Z_C(\tau,\bar{\tau})$ |
| $(p \neq 2)$ $H\eta H^T = 0 \bmod p$  $(p=2)$ $\mathrm{diag}(H\eta H^T) = 0 \bmod 4$ | self-orthogonal doubly-even | even | modular $T$ invariance |
| $[[n,0]]_p$ code s.t. $H\eta H^T = 0 \bmod p$ | self-dual | self-dual | modular $S$ invariance |

## 4.3 Example: CSS construction

Let us turn back to the partition functions of our Narain code CFTs. As in Proposition 4.4, the partition function is uniquely determined by the complete enumerator polynomial of a classical code $C$. We give some examples for Narain code CFTs focusing on the CSS construction described in Theorem 3.8.

Suppose that a CSS code has a check matrix $H_{(C,C^\perp)}$ with a classical $[n,k]_p$ code $C$ and its dual code $C^\perp$. Let $\mathcal{C}$ be a classical code generated by the matrix $G_H = H_{(C,C^\perp)}$. The complete enumerator polynomial of the code $\mathcal{C}$ is given by

$$W_{C,C^\perp}^{(\text{CSS})}(\{x_{ab}\}) := W_C(\{x_{ab}\}) = \sum_{c \in C, c' \in C^\perp} \prod_{(a,b) \in \mathbb{F}_p \times \mathbb{F}_p} x_{ab}^{\text{wt}_{ab}(c,c')}, \tag{105}$$

where $c = (c_1, \cdots, c_n) \in C$ and $c' = (c'_1, \cdots, c'_n) \in C^\perp$. Here, for $(a,b) \in \mathbb{F}_p \times \mathbb{F}_p$, we set

$$\text{wt}_{ab}(c,c') = \left| \{ j \in \{1, \cdots, n\} \,|\, c_j = a, \, c'_j = b \} \right|. \tag{106}$$

The complete enumerator polynomial of the CSS code is given in terms of a pair of classical codes $C$ and $C^\perp$. We point out that the complete enumerator polynomial can be understood as the 2-fold complete joint weight enumerator of classical codes $C$ and $C^\perp$.

Let us consider $r$ classical $[n,k_i]_p$ codes $C^{(i)}$ (possibly distinct) where $i = 1, 2, \cdots, r$, and define their product $\underline{C} = C^{(1)} \times \cdots \times C^{(r)}$. The $r$-fold complete joint weight enumerator for $\underline{C}$ is given by ( [14])

$$\mathcal{W}_{\underline{C}}(\{x_v\}) = \sum_{(c^{(1)}, \cdots, c^{(r)}) \in \underline{C}} \prod_{v \in \mathbb{F}_p^r} x_v^{\text{wt}_v(c^{(1)}, \cdots, c^{(r)})}, \tag{107}$$

where $v = (v_1, \cdots, v_r) \in \mathbb{F}_p^r$, $c^{(i)} = (c_1^{(i)}, \cdots, c_n^{(i)}) \in C^{(i)}$, and

$$\text{wt}_v(c^{(1)}, \cdots, c^{(r)}) = \left| \left\{ j \in \{1, \cdots, n\} \,\middle|\, c_j^{(i)} = v_i, \, i = 1, 2, \cdots, r \right\} \right|. \tag{108}$$

Note that this is a generalization of (106). If we set $r = 2$ and $(C^{(1)}, C^{(2)}) = (C, C^\perp)$, we arrive at the complete enumerator polynomial (105) for the CSS code. Then, we obtain

$$W_{C,C^\perp}^{(\text{CSS})}(\{x_{ab}\}) = \mathcal{W}_{\underline{C}}(\{x_{ab}\}),\tag{109}$$

where $\underline{C} = C \times C^\perp$. As dictated in Proposition 4.4, the partition functions of the Narain code CFTs are determined by the complete enumerator polynomial of the associated classical code. Therefore, the partition function for the CSS construction turns out to be

$$Z_{C,C^\perp}^{(\text{CSS})}(\tau, \bar{\tau}) = \frac{1}{|\eta(\tau)|^{2n}} W_{C,C^\perp}^{(\text{CSS})}(\{\psi_{ab}\}).\tag{110}$$

Our CSS construction can be applied to a CSS code based on a classical self-dual code $C = C^\perp$. Then, the 2-fold complete joint enumerator of $C$ and $C^\perp = C$ reduces to the genus-2 weight enumerator of $C$, which will be introduced in section 5 because it plays a significant role when averaging the partition functions over the CSS codes.

Finally, we give some examples of the CSS construction. Let us consider a trivial $[1, 0]_2$ code $C$ such that the generator matrix is $G_C = [0]$ and the check matrix is $H_C = [1]$. Then, the CSS construction gives the check matrix

$$\mathsf{H}_{(C,C^\perp)} = \left[\begin{array}{c|c} 1 & 0 \end{array}\right],\tag{111}$$

where we omit the row that comes from the generator matrix $G_C$ in (65) because it does not contribute to nontrivial generators. The stabilizer generator of the CSS code is the Pauli $X$. The corresponding complete enumerator polynomial is given by

$$W_{C,C^\perp}^{(\text{CSS})}(\{x_{ab}\}) = x_{00} + x_{10}.\tag{112}$$

Here, we have

$$\psi_{00} = \frac{\vartheta_3 \bar{\vartheta}_3 + \vartheta_4 \bar{\vartheta}_4}{2}, \qquad \psi_{01} = \psi_{10} = \frac{\vartheta_2 \bar{\vartheta}_2}{2}, \qquad \psi_{11} = \frac{\vartheta_3 \bar{\vartheta}_3 - \vartheta_4 \bar{\vartheta}_4}{2},\tag{113}$$

where $\vartheta_i$ ($i = 2, 3, 4$) are the Jacobi theta functions, $\vartheta_2(\tau) \equiv \sum_{n \in \mathbb{Z}} q^{\frac{1}{2}\left(n - \frac{1}{2}\right)^2}$, $\vartheta_3(\tau) \equiv \sum_{n \in \mathbb{Z}} q^{\frac{n^2}{2}}$, $\vartheta_4(\tau) \equiv \sum_{n \in \mathbb{Z}} (-1)^n q^{\frac{n^2}{2}}$, $q = e^{2\pi i \tau}$. Then, the partition function of the Narain code CFT becomes

$$Z_{C,C^\perp}^{(\text{CSS})}(\tau, \bar{\tau}) = \frac{\vartheta_2 \bar{\vartheta}_2 + \vartheta_3 \bar{\vartheta}_3 + \vartheta_4 \bar{\vartheta}_4}{2 |\eta(\tau)|^2}.\tag{114}$$

As another example, we consider the $[2, 1]_5$ self-dual code $C$ whose generator matrix is given by $G_C = [\,1\,2\,]$. The parity check matrix is also $H_C = [\,1\,2\,]$ because of self-duality. The check matrix of the corresponding CSS code is

$$\mathsf{H}_{(C,C)} = \left[\begin{array}{cc|cc} 1 & 2 & 0 & 0 \\ 0 & 0 & 1 & 2 \end{array}\right].\tag{115}$$

The complete enumerator polynomial is

$$\begin{aligned}
W_{C,C}^{(\text{CSS})}(\{x_{ab}\}) = {} & x_{00}^2 + x_{01}x_{02} + x_{01}x_{03} + x_{02}x_{04} + x_{03}x_{04} + x_{10}x_{20} + x_{13}x_{21} \\
& + x_{11}x_{22} + x_{14}x_{23} + x_{12}x_{24} + x_{10}x_{30} + x_{12}x_{31} + x_{14}x_{32} \\
& + x_{11}x_{33} + x_{13}x_{34} + x_{20}x_{40} + x_{30}x_{40} + x_{23}x_{41} + x_{32}x_{41} \\
& + x_{21}x_{42} + x_{34}x_{42} + x_{24}x_{43} + x_{31}x_{43} + x_{22}x_{44} + x_{33}x_{44}.
\end{aligned}\tag{116}$$

We obtain the partition function of the Narain code CFT

$$Z_{C,C}^{(\text{CSS})}(\tau, \bar{\tau}) = \frac{1}{|\eta(\tau)|^4} W_{C,C}^{(\text{CSS})}(\{\psi_{ab}\}),$$

(117)

where we substitute (95) to the variables $x_{ab}$.

# 5 Averaged partition function

Recently, the relation between an averaged theory over the whole Narain moduli and U(1) Chern-Simons theory with topological sum has been pointed out [15, 16]. In this section, we consider the averaged theory of the Narain code CFTs based on a class of CSS codes. Then, the average is a sum over the discrete points in the whole Narain moduli space. For a class of CSS codes defined by a single self-dual code $C$, we exactly compute the averaged partition functions of the associated Narain code CFTs. We will discuss the holographic implication of the averaged partition functions in section 6.

## 5.1 Higher-genus weight enumerator

We introduced the CSS construction for a pair $(C, C^\perp)$ with a classical code $C$ in section 4.3. In this section, we focus on a pair $(C, C)$ with self-dual codes $C$.

Let $\mathsf{H}_{(C,C)}$ be a check matrix of a CSS code that is constructed from a single self-dual code $C$ over $\mathbb{F}_p$ via (65). Suppose that $\mathcal{C}$ is a classical code generated by the matrix $G_\mathsf{H} = \mathsf{H}_{(C,C)}$. The complete enumerator polynomial of the classical code $\mathcal{C}$ is given by

$$W_{C,C}^{(\text{CSS})}(\{x_{ab}\}) = \sum_{(c,c') \in C^2} \prod_{(a,b) \in \mathbb{F}_p \times \mathbb{F}_p} x_{ab}^{\text{wt}_{ab}(c,c')},$$

(118)

where, for codewords $c = (c_1, \cdots, c_n) \in C$ and $c' = (c'_1, \cdots, c'_n) \in C$, we define

$$\text{wt}_{ab}(c, c') = \left| \{ j \in \{1, \cdots, n\} \mid c_j = a, \ c'_j = b \} \right|.$$

(119)

We aim to average the above complete enumerator polynomial over a classical self-dual code $C$ over $\mathbb{F}_p$ for fixed length $n$. Before taking the average, we interpret the complete enumerator polynomial of a quantum CSS code as a genus-2 weight enumerator of a classical self-dual code $C$. It is helpful for our task since the average of genus-$g$ weight enumerators over doubly-even self-dual codes was considered in [30, 31] for a binary case.

Let us introduce higher-genus weight enumerators of a classical code $C$ of length $n$. The genus-$g$ weight enumerator of a classical code $C$ over $\mathbb{F}_p$ is defined by

$$W_{g,C}(\{x_v\}) = \sum_{(c^{(1)}, \cdots, c^{(g)}) \in C^g} \prod_{v \in \mathbb{F}_p^g} x_v^{\text{wt}_v(c^{(1)}, \cdots, c^{(g)})},$$

(120)

where for $v = (v_1, \cdots, v_g) \in \mathbb{F}_p^g$ and $c^{(i)} = (c_1^{(i)}, \cdots, c_n^{(i)}) \in C$, the term $\text{wt}_v(c^{(1)}, \cdots, c^{(g)})$ is given by the following:

$$\text{wt}_v(c^{(1)}, \cdots, c^{(g)}) = \left| \left\{ j \in \{1, \cdots, n\} \mid c_j^{(i)} = v_i, i = 1, 2, \cdots, g \right\} \right|.$$

(121)

Note that $\text{wt}_v(c^{(1)}, \cdots, c^{(g)})$ is a generalization of (119) for $g \geq 2$ and reduces to (119) for $g = 2$. For $g = 1$, the above genus-$g$ weight enumerator becomes the usual complete enumerator polynomial of a classical code $C$.

The higher-genus weight enumerators are a reduced form of the complete joint weight enumerator $\mathcal{W}_{\underline{C}}(\{x_v\})$ introduced in section 4.3. Let us compare these definitions (107) and (120). The only difference is that the complete joint weight enumerator can deal with a product of different classical codes $\underline{C} = C^{(1)} \times \cdots \times C^{(r)}$. If we set $\underline{C} = C^r$, the $r$-fold complete joint weight enumerator reduces to the genus-$r$ weight enumerator of a classical code $C$:

$$\mathcal{W}_{\underline{C}}(\{x_v\}) = W_{r,C}(\{x_v\}). \tag{122}$$

Let us return to the complete enumerator polynomial (118) of the CSS code. In section 4.3, we pointed out the coincidence (109) between the 2-fold complete joint enumerator of $\underline{C} = C \times C^\perp$ and the complete enumerator polynomial of the classical code generated by the matrix $\mathsf{H}_{(C,C^\perp)}$: $W_{C,C^\perp}^{(\mathrm{CSS})}(\{x_{ab}\}) = \mathcal{W}_{\underline{C}}(\{x_{ab}\})$. Focusing on a self-dual code $C = C^\perp$, we get

$$W_{C,C}^{(\mathrm{CSS})}(\{x_{ab}\}) = \mathcal{W}_{\underline{C}}(\{x_{ab}\}), \tag{123}$$

where $\underline{C} = C \times C$. The relation (122) implies the following result:

$$W_{C,C}^{(\mathrm{CSS})}(\{x_{ab}\}) = W_{2,C}(\{x_{ab}\}). \tag{124}$$

Therefore, the complete weight enumerator of the CSS code can be understood as the genus-2 weight enumerator of the classical code $C$. Average of $W_{C,C}^{(\mathrm{CSS})}(\{x_{ab}\})$ over self-dual codes $C$ reduces to the sum of the genus-2 weight enumerator over classical self-dual codes.

Now we give an alternative expression of the higher-genus weight enumerator (120), which will be useful when we consider the average over the CSS codes in the next subsection.

Let $\mathfrak{c}$ be a tuple of $g$ elements $c^{(i)} \in \mathbb{F}_p^n$ ($i = 1, 2, \cdots, g$), denoted by $\mathfrak{c} = \left(c^{(1)}, \cdots, c^{(g)}\right)$. We associate $\mathfrak{c}$ to a tuple $A(\mathfrak{c}) = (e_v(\mathfrak{c}) \,|\, v \in \mathbb{F}_p^g)$ by

$$e_v(\mathfrak{c}) = \mathrm{wt}_v\left(c^{(1)}, \cdots, c^{(g)}\right). \tag{125}$$

To catch the meaning of this definition, consider $n = 4$, $g = 2$ and $p = 3$ case. Suppose we take two elements in $\mathbb{F}_{p=3}^{n=4}$ as

$$c^{(1)} = (1, 2, 0, 1), \qquad c^{(2)} = (0, 1, 2, 0). \tag{126}$$

The tuple $A(\mathfrak{c})$ can be read off from the four column vectors of the matrix whose rows are $c^{(i)}$:

$$\begin{bmatrix} c^{(1)} \\ c^{(2)} \end{bmatrix} = \left[ \begin{pmatrix} 1 \\ 0 \end{pmatrix} \begin{pmatrix} 2 \\ 1 \end{pmatrix} \begin{pmatrix} 0 \\ 2 \end{pmatrix} \begin{pmatrix} 1 \\ 0 \end{pmatrix} \right]. \tag{127}$$

It follows from the definitions (121) and (125) that $e_v(\mathfrak{c})$ counts the number of column vectors which match $v$. In this example, we have

$$e_{10}(\mathfrak{c}) = 2, \qquad e_{21}(\mathfrak{c}) = 1, \qquad e_{02}(\mathfrak{c}) = 1, \qquad e_{v \neq 10,21,02}(\mathfrak{c}) = 0. \tag{128}$$

Note that $e_v(\mathfrak{c})$ are a partition of $n$

$$n = \sum_{v \in \mathbb{F}_p^g} e_v(\mathfrak{c}), \tag{129}$$

which is verified in the above example.

With the tuple $A(\mathfrak{c})$, one can rewrite the genus-$g$ weight enumerator (120) as

$$W_{g,C}(\{x_v\}) = \sum_{\mathfrak{c} \in C^g} x^{A(\mathfrak{c})}, \tag{130}$$

where $x^{A(\mathfrak{c})} = \prod_{v \in \mathbb{F}_p^g} x_v^{e_v(\mathfrak{c})}$.

## 5.2 Average of higher-genus weight enumerator

We have found that the average of the complete enumerator polynomial over the CSS codes reduces to the sum of the genus-2 weight enumerator of self-dual codes. This section tackles more general problems: averaging the genus-$g$ weight enumerator over self-dual codes.

Let $\mathcal{M}_{n,p}$ be a set of classical self-dual codes $C \subset \mathbb{F}_p^n$ with $n$ and $p$ fixed:

$$\mathcal{M}_{n,p} = \{\text{self-dual codes over } \mathbb{F}_p \text{ of length } n\}. \tag{131}$$

The average of the genus-$g$ weight enumerators over a set of self-dual codes $\mathcal{M}_{n,p}$ is given by

$$E_{n,p}^{(g)}(\{x_\nu\}) = \frac{1}{|\mathcal{M}_{n,p}|} \sum_{C \in \mathcal{M}_{n,p}} W_{g,C}(\{x_\nu\}). \tag{132}$$

For doubly-even self-dual codes over $\mathbb{F}_2$ of length $n \in 8\mathbb{Z}$, these polynomials $E_{n,p}^{(g)}(\{x_\nu\})$ are called Eisenstein polynomials as being the counterpart of the Eisenstein series for lattices [31]. The genus-$g$ Eisenstein polynomials $E_{n,p}^{(g)}$ are given explicitly in [30,31].

In the following, we consider the averaged genus-$g$ weight enumerators over self-dual codes $C \subset \mathbb{F}_2^n$ and $C \subset \mathbb{F}_p^n$ for an odd prime $p$, respectively.

### 5.2.1 For $p = 2$

Let us introduce some notions needed to describe our statements.

**Type-I-admissible tuples**   We define a tuple $A = (e_\nu \,|\, \nu \in \mathbb{F}_2^g)$ where $e_\nu \in \mathbb{Z}_{\geq 0}$. We define the dimension of a tuple $A$ as the dimension of the vector space spanned by the vectors $(1\nu)$ satisfying $e_\nu > 0$:

$$\dim_2(A) = \dim_{\mathbb{F}_2} \langle \{(1\nu) \in \mathbb{F}_2^{g+1} \,|\, e_\nu > 0\} \rangle, \tag{133}$$

where $(1\nu) \in \mathbb{F}_2^{g+1}$ is the binary vector such that the first component is 1 and the remaining components are $\nu$, and $\langle \{a, b, c, \cdots\} \rangle$ denotes the vector space spanned by the vectors $a, b, c, \cdots$

We call $A$ a type-I-admissible tuple if a tuple $A$ is a partition of even $n$:

$$n = \sum_{\nu \in \mathbb{F}_2^g} e_\nu = 0 \qquad \mod 2, \tag{134}$$

and it satisfies

$$\sum_{\nu \in \mathbb{F}_2^g} e_\nu \left(\nu S_{\mathrm{d}} \nu^T\right) = 0 \mod 2, \qquad \sum_{\nu \in \mathbb{F}_2^g} e_\nu \left(\nu S_{\mathrm{nd}} \nu^T\right) = 0 \mod 4, \tag{135}$$

where $\nu = (\nu_1, \cdots, \nu_g) \in \mathbb{F}_2^g$ for all integral diagonal $g \times g$ matrices $S_{\mathrm{d}}$ and all integral symmetric $g \times g$ matrices $S_{\mathrm{nd}}$ with 0s in diagonal elements.

Let us illustrate the above definition of the dimension of a tuple by an example. Consider the genus-two ($g = 2$) and $n = 8$ case where $\nu$ is a two-dimensional binary vector, $\nu \in \{00, 01, 10, 11\} = \mathbb{F}_2^2$. Let us take a tuple $A = (e_\nu \,|\, \nu \in \mathbb{F}_2^2) = (e_{00}, e_{01}, e_{10}, e_{11})$ such that

$$e_{00} = 2, \qquad e_{01} = 4, \qquad e_{10} = 2, \qquad e_{11} = 0. \tag{136}$$

Then its dimension is given by

$$\dim_2(A) = \dim_{\mathbb{F}_2} \langle \{(1\nu) \in \mathbb{F}_2^3 \,|\, e_\nu > 0\} \rangle = \dim_{\mathbb{F}_2} \langle \{100, 101, 110\} \rangle = 3. \tag{137}$$

To see if the tuple $A$ is type-I-admissible, we examine the conditions (135) for $2 \times 2$ matrices of the forms:

$$S_{\mathrm{d}} = \begin{pmatrix} a & 0 \\ 0 & b \end{pmatrix}, \qquad S_{\mathrm{nd}} = \begin{pmatrix} 0 & c \\ c & 0 \end{pmatrix}, \qquad (a, b, c \in \mathbb{Z}). \tag{138}$$

Then, the conditions (135) become

$$\sum_{v_1, v_2 \in \mathbb{F}_2} e_v (a\, v_1^2 + b\, v_2^2) = a\,(e_{10} + e_{11}) + b\,(e_{01} + e_{11}) = 0 \mod 2\,,$$

$$2 \sum_{v_1, v_2 \in \mathbb{F}_2} e_v\, c\, v_1\, v_2 = 2\, c\, e_{11} = 0 \mod 4\,. \tag{139}$$

For the tuple (136), these equations hold for any integer $a, b, c$. Thus, the tuple $A$ is type-I-admissible in this example.

**Self-orthogonal codes and type-I-admissible tuples** For a tuple of $g$ elements $\mathfrak{c} = (c^{(1)}, \cdots, c^{(g)}) \in (\mathbb{F}_2^n)^g$, we define $\mathfrak{C}$ as the $[n, s(\mathfrak{c})]_2$ code generated by $\mathbf{1}_n$ and $c^{(1)}, \cdots, c^{(g)}$ where $s(\mathfrak{c})$ is the dimension of the code. On the other hand, we associate $\mathfrak{c}$ to a tuple $A(\mathfrak{c}) = (e_v(\mathfrak{c}) | v \in \mathbb{F}_2^g)$ as in (125). There is a simple relation between the tuple $A(\mathfrak{c})$ and the dimension $s(\mathfrak{c})$:

$$\dim_2(A(\mathfrak{c})) = s(\mathfrak{c}). \tag{140}$$

To show this equality, note that the dimension of a code generated by $\mathbf{1}_n$ and $c^{(1)}, \cdots, c^{(g)} \in \mathbb{F}_2^n$ is given by the following:

$$s(\mathfrak{c}) = \mathrm{rank} \begin{bmatrix} 1 & 1 & \cdots & 1 \\ c_1^{(1)} & c_2^{(1)} & \cdots & c_n^{(1)} \\ \vdots & \vdots & & \vdots \\ c_1^{(g)} & c_2^{(g)} & \cdots & c_n^{(g)} \end{bmatrix}. \tag{141}$$

Elementary column operations reduce the right hand side to the dimension of the vector space spanned by the vectors $\{(1v) \in \mathbb{F}_2^{g+1} \mid e_v > 0\}$. Therefore, we arrive at the relation $s(\mathfrak{c}) = \dim_2(A(\mathfrak{c}))$.

Now we state an important relation between the code $\mathfrak{C}$ and the tuple $A(\mathfrak{c})$ which will play a key role in deriving the averaged weight enumerator:

**Proposition 5.1**
Let $\mathfrak{c}$ be a tuple of $g$ elements $\mathfrak{c} = (c^{(1)}, \cdots, c^{(g)}) \in (\mathbb{F}_2^n)^g$. Then, the code $\mathfrak{C}$ generated by $\mathbf{1}_n$ and $\mathfrak{c}$ is self-orthogonal code of length $n$ if and only if the associated tuple $A(\mathfrak{c})$ is type-I-admissible.

*Proof.* Assume that $\mathfrak{C}$ is self-orthogonal. Let $c^{(i)}$ ($i = 1, 2, \cdots, g$) be elements in the tuple $\mathfrak{c}$. The code $\mathfrak{C}$ generated by $\mathbf{1}_n$ and $c^{(i)}$ ($i = 1, 2, \cdots, g$) is self-orthogonal if and only if $\mathbf{1}_n \cdot \mathbf{1}_n = 0$ mod 2, $\mathbf{1}_n \cdot c^{(i)} = c^{(i)} \cdot c^{(i)} = 0$ mod 2, and $c^{(i)} \cdot c^{(j)} = 0$ mod 2 for $i \neq j$. The first condition implies $n = \sum_{v \in \mathbb{F}_2^g} e_v(\mathfrak{c}) = 0$ mod 2. Let us denote a binary vector $v = (v_1, \cdots, v_g) \in \mathbb{F}_2^g$ where $v_i \in \mathbb{F}_2$ for convenience. Then

$$c^{(i)} \cdot c^{(i)} = \sum_{v_i = 1, v_{j \neq i} \in \mathbb{F}_2} e_{v_1 \cdots v_g}(\mathfrak{c}) = \sum_{v_i \in \mathbb{F}_2} v_i^2 \sum_{v_{j \neq i} \in \mathbb{F}_2} e_{v_1 \cdots v_g}(\mathfrak{c})$$

$$= \sum_{v_1, \cdots, v_g \in \mathbb{F}_2} e_{v_1 \cdots v_g}(\mathfrak{c})\, v_i^2 = \sum_{v \in \mathbb{F}_2^g} e_v(\mathfrak{c}) \left( v\, S_{\mathrm{d}}^{(i)}\, v^T \right), \tag{142}$$

where $S_{\mathrm{d}}^{(i)}$ is the diagonal $g \times g$ matrix with 1 at the $(i,i)$-th position and 0s elsewhere. The condition $c^{(i)} \cdot c^{(i)} = 0 \bmod 2$ for $i = 1, 2, \cdots, g$ implies

$$\sum_{v \in \mathbb{F}_2^g} e_v(\mathfrak{c}) \left(v S_{\mathrm{d}} v^T\right) = 0 \qquad \bmod 2\,, \tag{143}$$

for all integral diagonal $g \times g$ matrices $S_{\mathrm{d}}$. Also, we have for $i \neq j$

$$c^{(i)} \cdot c^{(j)} = \sum_{v_1, \cdots, v_g \in \mathbb{F}_2} e_{v_1 \cdots v_g}(\mathfrak{c}) v_i v_j = \frac{1}{2} \sum_{v \in \mathbb{F}_2^g} e_v(\mathfrak{c}) \left(v S_{\mathrm{nd}}^{(i,j)} v^T\right), \tag{144}$$

where $S_{\mathrm{nd}}^{(i,j)}$ is the symmetric $g \times g$ matrix with 1 at the $(i,j)$-th and $(j,i)$-th positions, and 0s elsewhere. Then the other condition $c^{(i)} \cdot c^{(j)} = 0 \bmod 2$ becomes

$$\sum_{v \in \mathbb{F}_2^g} e_v(\mathfrak{c}) \left(v S_{\mathrm{nd}} v^T\right) = 0 \qquad \bmod 4\,, \tag{145}$$

for all integral symmetric $g \times g$ matrices $S_{\mathrm{nd}}$ with 0s in diagonal elements. Therefore, self-orthogonality for $\mathfrak{C}$ means that a tuple $A(\mathfrak{c})$ is type-I-admissible. If $A(\mathfrak{c}) = (e_v(\mathfrak{c}) \,|\, v \in \mathbb{F}_2^g)$ is type-I-admissible, we can trace the above discussion backwards. $\qquad\square$

**Main theorem and its proof**   The following theorem gives the average of genus-$g$ weight enumerators for self-dual codes over $\mathbb{F}_2$. For doubly-even self-dual codes, it was shown in [30, 31]. To our best knowledge, however, the formula for self-dual codes over $\mathbb{F}_2$ has not been stated explicitly in literature.

**Theorem 5.2**

Let $\mathcal{M}_{n,2}$ be a set of classical self-dual codes over $\mathbb{F}_2$ of length $n \in 2\mathbb{Z}$. Then the average of genus-$g$ weight enumerators is given by

$$
\begin{aligned}
E_{n,2}^{(g)}(\{x_v\}) &:= \frac{1}{|\mathcal{M}_{n,2}|} \sum_{C \in \mathcal{M}_{n,2}} W_{g,C}(\{x_v\}) \\
&= \sum_A \frac{1}{\left(2^{\frac{n}{2}-1}+1\right) \cdots \left(2^{\frac{n}{2}-\dim_2(A)+1}+1\right)} \binom{n}{A} x^A\,,
\end{aligned}
\tag{146}
$$

where the sum is extended over all type-I-admissible tuples $A = (e_v \,|\, v \in \mathbb{F}_2^g)$. We denote $x^A = \prod_{v \in \mathbb{F}_2^g} x_v^{e_v}$ and

$$\binom{n}{A} = \frac{n!}{\prod_{v \in \mathbb{F}_2^g} e_v!}\,. \tag{147}$$

*Proof.* We prove the theorem following [30] where the averaged genus-$g$ weight enumerator over doubly-even self-dual codes over $\mathbb{F}_2$ is given.

First, we use the tuple representation (130) of $W_{g,C}(\{x_v\})$ to rewrite the averaged weight enumerator:

$$
\begin{aligned}
E_{n,2}^{(g)}(\{x_v\}) &= \frac{1}{|\mathcal{M}_{n,2}|} \sum_{C \in \mathcal{M}_{n,2}} W_{g,C}(\{x_v\}) \\
&\underset{(130)}{=} \frac{1}{|\mathcal{M}_{n,2}|} \sum_{C \in \mathcal{M}_{n,2}} \sum_{\mathfrak{c} \in C^g} x^{A(\mathfrak{c})} \\
&= \frac{1}{|\mathcal{M}_{n,2}|} \sum_{\mathfrak{c} \in \mathbb{F}_2^{ng}} \left|\{C \in \mathcal{M}_{n,2} \text{ with } \mathfrak{c} \in C^g\}\right| x^{A(\mathfrak{c})}\,,
\end{aligned}
\tag{148}
$$

where $\left|\{C \in \mathcal{M}_{n,2} \text{ with } \mathfrak{c} \in C^g\}\right|$ is the number of binary self-dual codes of length $n$ such that $C^g$ contains a tuple $\mathfrak{c}$.

The number of binary self-dual $[n, n/2]_2$ codes, which contain a self-orthogonal $[n,s]_2$ code including $\mathbf{1}_n \in \mathbb{F}_2^n$, is [56, Theorem 2.1]

$$\prod_{i=1}^{\frac{n}{2}-s}(2^i+1). \tag{149}$$

Note that self-dual codes contain only self-orthogonal codes because any subspaces of self-dual codes are self-orthogonal.

Since all binary self-dual codes contain the all-ones vector $\mathbf{1}_n$, self-dual codes containing $c^{(i)}$ ($i = 1, 2, \cdots, g$) always contain the code $\mathfrak{C}$. It is obvious that self-dual codes contain the codewords $c^{(i)}$ ($i = 1, 2, \cdots, g$) if they contain the code $\mathfrak{C}$. Hence, we get

$$\left|\{C \in \mathcal{M}_{n,2} \text{ with } \mathfrak{c} \in C^g\}\right| = \left|\{C \in \mathcal{M}_{n,2} \text{ with } \mathfrak{C} \subset C\}\right|. \tag{150}$$

Using the enumeration (149), we count the number of self-dual codes $C$ such that $\mathfrak{c} \in C^g$ as follows

$$\left|\{C \in \mathcal{M}_{n,2}, \text{ with } \mathfrak{c} \in C^g\}\right| = \begin{cases} \displaystyle\prod_{i=1}^{\frac{n}{2}-s(\mathfrak{c})}(2^i+1), & \text{if } \mathfrak{C} \text{ self-orthogonal,} \\ 0, & \text{otherwise.} \end{cases} \tag{151}$$

Consider the case with $s = 1$ in (149). The formula returns the number of self-dual codes containing a self-orthogonal $[n,1]_2$ code $\mathfrak{C}$ that contains $\mathbf{1}_n \in \mathbb{F}_2^n$. All binary self-dual codes contain the $[n,1]_2$ code that consists of the all-zeros vector and the all-ones vector. Therefore, (149) gives the number of whole binary self-dual codes enumerated in [57]

$$|\mathcal{M}_{n,2}| = \prod_{i=1}^{\frac{n}{2}-1}(2^i+1). \tag{152}$$

Therefore, the averaged genus-$g$ weight enumerator can be written as

$$\begin{aligned} E_{n,2}^{(g)}(\{x_\nu\}) &= \frac{1}{|\mathcal{M}_{n,2}|} \sum_{\mathfrak{c} \in \mathbb{F}_2^{ng}} \left|\{C \in \mathcal{M}_{n,2} \text{ with } \mathfrak{c} \in C^g\}\right| x^{A(\mathfrak{c})} \\ &\underset{\substack{(151) \\ (140)}}{=} \sum_{\mathfrak{C} \subset \mathfrak{C}^\perp} \frac{1}{\left(2^{\frac{n}{2}-1}+1\right) \cdots \left(2^{\frac{n}{2}-\dim_2(A(\mathfrak{c}))+1}+1\right)} x^{A(\mathfrak{c})} \\ &= \sum_A \frac{1}{\left(2^{\frac{n}{2}-1}+1\right) \cdots \left(2^{\frac{n}{2}-\dim_2(A)+1}+1\right)} \binom{n}{A} x^A. \end{aligned} \tag{153}$$

In the second line, the sum is extended over tuples $\mathfrak{c}$ such that $\mathfrak{C}$ is self-orthogonal. In the last line, we take the sum over type-I-admissible tuples $A$. The last line follows from Proposition 5.1 and the fact that, for a type-I-admissible tuple $A = (e_\nu \mid \nu \in \mathbb{F}_2^g)$, there are $\binom{n}{A}$ tuples $\mathfrak{c}$ which are different only in the order of coordinates. $\qquad\square$

For $g = 1$, the genus-$g$ weight enumerator reduces to the usual complete enumerator polynomial. Then its average gives a well-known formula for the averaged enumerator polynomial over self-dual codes $C \subset \mathbb{F}_2^n$ (see for example p.329 in [53]):

$$E_{n,2}^{(1)}(\{x_0, x_1\}) = x_0^n + x_1^n + \frac{1}{2^{\frac{n}{2}-1}+1} \sum_{i=1}^{\frac{n}{2}-1} \binom{n}{2i} x_0^{2i} x_1^{n-2i}. \tag{154}$$

### 5.2.2 For odd prime $p \neq 2$

**$p$-admissible tuples**  For a tuple $A = (e_v \mid v \in \mathbb{F}_p^g)$, we define the dimension of the tuple

$$\dim_p(A) = \dim_{\mathbb{F}_p} \langle \{ v \in \mathbb{F}_p^g \mid e_v > 0 \} \rangle . \tag{155}$$

We call a tuple $A$ as $p$-admissible if

$$n = \sum_{v \in \mathbb{F}_p^g} e_v = \begin{cases} 0 \mod 2, & (p = 1 \mod 4), \\ 0 \mod 4, & (p = 3 \mod 4), \end{cases} \tag{156}$$

and

$$\sum_{v \in \mathbb{F}_p^g} e_v \left( v S v^T \right) = 0 \qquad \mod p, \tag{157}$$

for all integral symmetric $g \times g$ matrices $S$ where $v = (v_1, \cdots, v_g) \in \mathbb{F}_p^g$.

**Self-orthogonal codes and $p$-admissible tuples**  Let us take a tuple of $g$ elements $\mathfrak{c} = (c^{(1)}, \cdots, c^{(g)}) \in (\mathbb{F}_p^n)^g$. For each tuple $\mathfrak{c}$, we define $\mathfrak{C}$ as the $[n, s(\mathfrak{c})]_p$ code generated by $c^{(1)}, \cdots, c^{(g)}$ where $s(\mathfrak{c})$ is the dimension of the code. On the other hand, we associate $\mathfrak{c}$ to a tuple $A(\mathfrak{c}) = (e_v(\mathfrak{c}) \mid v \in \mathbb{F}_p^g)$ as in (125). There is a relation $\dim_p(A(\mathfrak{c})) = s(\mathfrak{c})$ because the dimension of the code $\mathfrak{C}$ generated by $c^{(1)}, \cdots, c^{(g)}$ is given by

$$s(\mathfrak{c}) = \mathrm{rank} \begin{bmatrix} c_1^{(1)} & c_2^{(1)} & \cdots & c_n^{(1)} \\ \vdots & \vdots & & \vdots \\ c_1^{(g)} & c_2^{(g)} & \cdots & c_n^{(g)} \end{bmatrix}, \tag{158}$$

which reduces to $s(\mathfrak{c}) = \dim_p(A(\mathfrak{c}))$ through the elementary column operations.

**Proposition 5.3**
Let $\mathfrak{c}$ be a tuple of $g$ elements $\mathfrak{c} = (c^{(1)}, \cdots, c^{(g)}) \in (\mathbb{F}_p^n)^g$. Then, the code $\mathfrak{C}$ generated by $\mathfrak{c}$ is self-orthogonal code of length $n$ if and only if the associated tuple $A(\mathfrak{c})$ is $p$-admissible.

*Proof.* Let $c^{(i)}$ ($i = 1, \cdots, g$) be elements in a tuple $\mathfrak{c}$. Note that $\mathfrak{C}$ is self-orthogonal if and only if $c^{(i)} \cdot c^{(i)} = 0 \mod p$, and $c^{(i)} \cdot c^{(j)} = 0 \mod p$ for $i \neq j$. Let us denote $v = (v_1, \cdots, v_g) \in \mathbb{F}_p^g$ for convenience. Then we have

$$c^{(i)} \cdot c^{(i)} = \sum_{v_i = 1, v_{j \neq i} \in \mathbb{F}_p} e_{v_1 \cdots v_g} = \sum_{v_i \in \mathbb{F}_p} v_i^2 \sum_{v_{j \neq i} \in \mathbb{F}_p} e_{v_1 \cdots v_g}(\mathfrak{c})$$
$$= \sum_{v_1, \cdots, v_g \in \mathbb{F}_p} e_{v_1 \cdots v_g}(\mathfrak{c}) v_i^2 = \sum_{v \in \mathbb{F}_p^g} e_v(\mathfrak{c}) \left( v S_{\mathrm{d}} v^T \right), \tag{159}$$

where $S_{\mathrm{d}}$ is the diagonal $g \times g$ matrix with 1 at the $(i, i)$-th position and 0s elsewhere. Therefore, $c^{(i)} \cdot c^{(i)} = 0 \mod p$ if and only if $\sum_{v \in \mathbb{F}_p^g} e_v(\mathfrak{c}) \left( v S_{\mathrm{d}} v^T \right) = 0 \mod p$. Also, we have for $i \neq j$

$$c^{(i)} \cdot c^{(j)} = \sum_{v_1, \cdots, v_g \in \mathbb{F}_p} e_{v_1 \cdots v_g}(\mathfrak{c}) v_i v_j = \frac{1}{2} \sum_{v \in \mathbb{F}_p^g} e_v(\mathfrak{c}) \left( v S_{\mathrm{nd}} v^T \right), \tag{160}$$

where $S_{\mathrm{nd}}$ is the symmetric $g \times g$ matrix with 1 at the $(i, j)$-th and $(j, i)$-th position, and 0s elsewhere. Since we have $c^{(i)} \cdot c^{(j)} \in \mathbb{Z}$, $c^{(i)} \cdot c^{(j)} = 0 \mod p$ if and only if $\sum_{v \in \mathbb{F}_p^g} e_v(\mathfrak{c}) \left( v S_{\mathrm{nd}} v^T \right) = 0$ mod $p$. Hence, a code $\mathfrak{C}$ is self-orthogonal if and only if $\sum_{v \in \mathbb{F}_p^g} e_v(\mathfrak{c}) \left( v S v^T \right) = 0 \mod p$ for all integral symmetric $g \times g$ matrices $S$. Let us consider the other condition (156) for a $p$-admissible tuple. Since we have $n = \sum_{v \in \mathbb{F}_p^g} e_v(\mathfrak{c})$, (156) holds automatically by the assumption in Theorem 5.4. Therefore, a code $\mathfrak{C}$ is self-orthogonal if and only if $A(\mathfrak{c})$ is $p$-admissible. $\square$

**Main theorem and its proof**

**Theorem 5.4**

Let $\mathcal{M}_{n,p}$ be a set of classical self-dual codes $C \subset \mathbb{F}_p^n$ for an odd prime $p$. (Then $n \in 2\mathbb{Z}$ for $p = 1 \mod 4$ and $n \in 4\mathbb{Z}$ for $p = 3 \mod 4$.) The average of genus-$g$ weight enumerators is given by

$$E_{n,p}^{(g)}(\{x_v\}) := \frac{1}{|\mathcal{M}_{n,p}|} \sum_{C \in \mathcal{M}_{n,p}} W_{g,C}(\{x_v\}) = \sum_A \frac{1}{\left(p^{\frac{n}{2}-1}+1\right)\cdots\left(p^{\frac{n}{2}-\dim_p(A)}+1\right)} \binom{n}{A} x^A, \quad (161)$$

where we take the sum over all $p$-admissible tuples. We denote $x^A = \prod_{v \in \mathbb{F}_p^g} x_v^{e_v}$ and

$$\binom{n}{A} = \frac{n!}{\prod_{v \in \mathbb{F}_p^g} e_v!}. \quad (162)$$

*Proof.* The proof is similar to the case with $p = 2$. The number of $p$-ary self-dual $[n, n/2]_p$ codes that contain a self-orthogonal $[n, s]_p$ code is [58]

$$2 \prod_{i=1}^{\frac{n}{2}-s-1} (p^i + 1). \quad (163)$$

Then the number of $p$-ary self-dual codes that contain the code $\mathfrak{C}$ is given by

$$|\{C \in \mathcal{M}_{n,p} \text{ with } \mathfrak{c} \in C^g\}| = \begin{cases} 2 \displaystyle\prod_{i=1}^{\frac{n}{2}-s(\mathfrak{c})-1} (p^i + 1), & \text{if } \mathfrak{C} \text{ self-orthogonal}, \\ 0, & \text{otherwise.} \end{cases} \quad (164)$$

For $s = 0$, (163) reduces to the number of $p$-ary self-dual codes of length $n$ enumerated in [57]

$$|\mathcal{M}_{n,p}| = 2 \prod_{i=1}^{\frac{n}{2}-1} (p^i + 1). \quad (165)$$

The averaged genus-$g$ weight enumerator is

$$\begin{aligned}
E_{n,p}^{(g)}(\{x_v\}) &= \frac{1}{|\mathcal{M}_{n,p}|} \sum_{C \in \mathcal{M}_{n,p}} W_{g,C}(\{x_v\}) \\
&= \frac{1}{|\mathcal{M}_{n,p}|} \sum_{C \in \mathcal{M}_{n,p}} \sum_{\mathfrak{c} \in C^g} x^{A(\mathfrak{c})} \\
&= \frac{1}{|\mathcal{M}_{n,p}|} \sum_{\mathfrak{c} \in \mathbb{F}_p^g} |\{C \in \mathcal{M}_{n,p} \text{ with } \mathfrak{c} \in C^g\}| x^{A(\mathfrak{c})} \\
&= \sum_{\mathfrak{C} \subset \mathfrak{C}^\perp} \frac{1}{\left(p^{\frac{n}{2}-1}+1\right)\cdots\left(p^{\frac{n}{2}-\dim_p(A(\mathfrak{c}))}+1\right)} x^{A(\mathfrak{c})} \\
&= \sum_A \frac{1}{\left(p^{\frac{n}{2}-1}+1\right)\cdots\left(p^{\frac{n}{2}-\dim_p(A)}+1\right)} \binom{n}{A} x^A.
\end{aligned} \quad (166)$$

In the fourth line, the sum is taken over tuples $\mathfrak{c}$ such that $\mathfrak{C}$ is self-orthogonal. In the last line, we take the sum over $p$-admissible tuples $A$. We obtain the last line because, for a $p$-admissible tuple $A = (e_v \mid v \in \mathbb{F}_p^g)$, there are $\binom{n}{A}$ tuples $\mathfrak{c}$ which are different only in the order of coordinates. $\square$

## 5.3 Averaging over CSS codes

Let us go back to the complete enumerator polynomial of a CSS code whose check matrix is given by $H_{(C,C)}$. As discussed in section 5.1, the complete enumerator polynomial can be written as the genus-2 weight enumerator of the associated classical self-dual code $C$:

$$W_{C,C}^{(CSS)}(\{x_{ab}\}) = W_{2,C}(\{x_{ab}\}). \tag{167}$$

Therefore, the average of the complete enumerator polynomials over a set of self-dual codes $\mathcal{M}_{n,p}$ reduces to the averaged genus-2 weight enumerator over $\mathcal{M}_{n,p}$:

$$
\begin{aligned}
\overline{W}_{n,p}^{(CSS)}(\{x_{ab}\}) &= \frac{1}{|\mathcal{M}_{n,p}|} \sum_{C \in \mathcal{M}_{n,p}} W_{C,C}^{(CSS)}(\{x_{ab}\}) \\
&= \frac{1}{|\mathcal{M}_{n,p}|} \sum_{C \in \mathcal{M}_{n,p}} W_{2,C}(\{x_{ab}\}) \\
&= E_{n,p}^{(2)}(\{x_{ab}\}) \\
&= \begin{cases} \displaystyle\sum_A \frac{1}{\left(2^{\frac{n}{2}-1}+1\right)\cdots\left(2^{\frac{n}{2}-\dim_2(A)+1}+1\right)} \binom{n}{A} x^A, & \text{if } p=2, \\ \displaystyle\sum_A \frac{1}{\left(p^{\frac{n}{2}-1}+1\right)\cdots\left(p^{\frac{n}{2}-\dim_p(A)+1}+1\right)} \binom{n}{A} x^A, & \text{if } p \text{ odd prime.} \end{cases}
\end{aligned}
\tag{168}
$$

Let us evaluate $\overline{W}_{n,p}^{(CSS)}(\{x_{ab}\})$ in the large-$n$ limit. To approximate the sums by integrals, we define variables

$$z_{ab} := \frac{e_{ab}}{n}. \tag{169}$$

Since the tuple $A = (e_{ab}|a, b \in \mathbb{F}_p)$ is a partition of $n$, we have $z_{ab} \geq 0$ and $\sum_{a,b} z_{ab} = 1$. In the large-$n$ limit the sums over $A$ become $(p^2-1)$-dimensional integrals over $\left(z_{ab}|(a,b)\neq(0,0)\right)$ in the region defined by $z_{ab} \geq 0$ and $\sum_{(a,b)\neq(0,0)} z_{ab} \leq 1$. When $p = 2$, for the tuple $A$ to be type-I admissible, $e_{ab}$ must satisfy the conditions

$$e_{01} = e_{10} = e_{11} = 0 \qquad \mod 2, \tag{170}$$

which follow from (139). When $p$ is an odd prime integer, for $A$ to be $p$-admissible, $e_{ab}$ must obey

$$\sum_{a,b\in\mathbb{F}_p} a^2 e_{ab} = \sum_{a,b\in\mathbb{F}_p} b^2 e_{ab} = \sum_{a,b\in\mathbb{F}_p} a\, b\, e_{ab} = 0 \qquad \mod p, \tag{171}$$

as follow from (156). Given generic values of the variables $(z_{ab}|a \neq 0, b \neq 0)$, the condition (170) or (171) reduces the number of allowed values of $(z_{01}, z_{10}, z_{11})$ by $p^3$ in either case. We also note that for generic $A$, the dimension defined by (133) and (155) is $\dim_p(A) = 3$ for $p = 2$ and $\dim_p(A) = 2$ for odd prime $p$. In both cases, in the large-$n$ limit, (168) is approximated by a $(p^2-1)$-dimensional integral over the region defined above

$$\overline{W}_{n,p}^{(CSS)}(\{x_{ab}\}) \simeq p^{-n}\left(\frac{n}{2\pi}\right)^{\frac{p^2-1}{2}} \int \left(\prod_{(a,b)\neq(0,0)} dz_{ab}\right) \prod_{a,b} \left(\frac{x_{ab}}{z_{ab}}\right)^{nz_{ab}} z_{ab}^{-\frac{1}{2}}, \tag{172}$$

where we used Stirling's formula $n! = (2\pi n)^{1/2}(n/e)^n(1+\mathcal{O}(n^{-1}))$. In Appendix B we evaluate the integral (172) by the saddle point method. We find that

$$\overline{W}_{n,p}^{(CSS)}(\{x_{ab}\}) = p^{-n}\left(\sum_{a,b} x_{ab}\right)^n (1+\mathcal{O}(n^{-1})). \tag{173}$$

To explore the density of states, let us simplify the averaged partition function by fixing the torus moduli $\tau = i\tau_2 = i\beta/2\pi$ ($\tau_1 = 0$). Then we have $q = \bar{q} = e^{-\beta}$ and $\psi_{ab} = \psi_a \psi_b$ where

$$\psi_a(i\tau_2) = \sum_{k \in \mathbb{Z}} q^{\frac{p}{2}\left(\frac{a}{p}+k\right)^2}. \tag{174}$$

Due to the relation (117) between the partition function and the enumerator polynomial, the averaged partition function reduces, for large $n$, to

$$
\begin{aligned}
\overline{Z}_{n,p}^{(\text{CSS})}(i\tau_2) &\simeq \frac{1}{p^n \, |\eta(i\tau_2)|^{2n}} \left( \sum_{a \in \mathbb{F}_p} \psi_a(i\tau_2) \right)^{2n} \\
&= \frac{1}{p^n \, |\eta(i\tau_2)|^{2n}} \left( \sum_{k \in \mathbb{Z}} e^{-\pi\tau_2 \frac{k^2}{p}} \right)^{2n} \\
&= \frac{\vartheta_3(i\tau_2/p)^{2n}}{p^n |\eta(i\tau_2)|^{2n}}.
\end{aligned}
\tag{175}
$$

The above partition function exactly agrees with the averaged partition function over the B-form codes in the large-$n$ limit, which was conjectured in [28].

# 6 Discussion

In this paper, we constructed a class of Narain code CFTs from $p$-ary qudit stabilizer codes for a prime $p$. Our construction was based on two fundamental relations: one between qudit codes to classical codes [8–10] and the other between classical codes and Lorentzian lattice [11]. The former is actually not limited to the case we considered but holds between $p^m$-ary stabilizer codes and self-orthogonal classical codes over $\mathbb{F}_{p^{2m}}$ for arbitrary integer $m \geq 1$ [10]. The latter relation is also likely to hold true for a broader class of classical codes [11]. Thus, we speculate that there is a class of Narain code CFTs associated with $p^m$-ary stabilizer codes for any integer $m \geq 1$.

In section 5, we considered the CSS codes as a special class of qudit codes and examined the averaged theory over the corresponding Narain code CFTs along the line of [15, 16]. We showed that the averaged partition function over the CSS codes takes the same form as the conjectured form of the partition function averaged over the B-form codes in the large-$n$ limit. Our definition of the averaged partition function is different from theirs as we take the average over all CSS codes associated with self-dual classical codes including equivalent ones while their averaging is over inequivalent qudit stabilizer codes. In the large-$n$ limit, this difference may be ignorable. Also our result implies that the CSS codes sample a typical set of quantum codes in this limit.

In section 5.3, we calculated the averaged partition function over the CSS codes (175). By noting that $|\eta(i\tau_2)|^{2n}$ in the denominator accounts for the descendant contributions, the density of primary states $\rho(\Delta)$ for the averaged code CFT can be read off from $\overline{Z}_{n,p}^{(\text{CSS})}(i\tau_2)$ with $\tau_2 = \frac{\beta}{2\pi}$ as

$$\frac{\vartheta_3(i\beta/2\pi p)^{2n}}{p^n} = \int_0^\infty d\Delta \, e^{-\beta \Delta} \rho(\Delta). \tag{176}$$

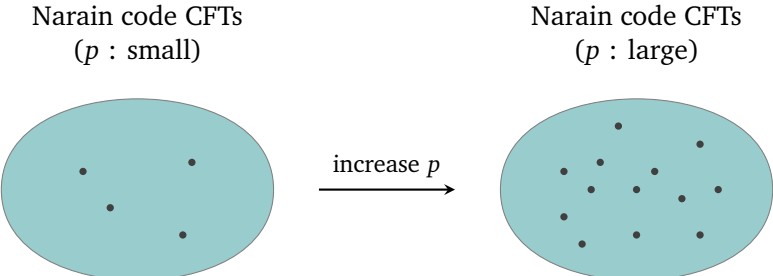

Figure 2: The discrete subset of Narain code CFTs constructed from the CSS codes (the black dots) in the whole Narain moduli space (the green region), which depends on a prime number $p$. For small $p$, the corresponding Narain code CFTs make a relatively small subset (black dots on the left). On the other hand, for large $p$, the number of the Narain code CFTs grows (black dots on the right) and we expect that the averaged theory over the ensemble resembles the one over the whole Narain moduli.

The asymptotic form of $\rho(\Delta)$ in $\Delta \to \infty$ is[5]

$$\rho(\Delta) \simeq \frac{(2\pi)^n \Delta^{n-1}}{\Gamma(n)} \,. \tag{177}$$

Numerical experiments suggest that this asymptotic form is valid for $\Delta \gtrsim \frac{1}{p} \log n$ and is likely to be exact when $p$ is large enough compared to $\log n$.[6] The density of states (177) is the same as the one for the averaged CFT of central charge $c = n$ over the whole Narain moduli, which is shown to have a spectral gap $\Delta = \frac{c}{2\pi e}$ in the large-$c$ limit [16]. We expect that our averaged Narain code CFT over the CSS codes has the same spectral gap in the large-$n$ limit for a large prime integer $p$ and that there exists a Narain CSS code CFT with the spectral gap $\Delta = \frac{c}{2\pi e}$.

In recent studies, ensemble averaging is seen as a key to understanding holographic duality [60]. The average of Narain CFTs has a large spectral gap and has been proposed to have a holographic interpretation in terms of an abelian Chern-Simons theory [15, 16]. We expect that our averaged theory over the CSS codes also has a large spectral gap, and may have a dual gravity description in the large-$n$ limit. Note that our ensemble average depends on the choice of a prime number $p$. We conjecture that each ensemble has a different gravity description as in [17, 18, 21, 22, 27–29]. In our case, the size of the ensemble increases for larger $p$ as in (165) and we expect the averaged theory tends to the one over the whole Narain moduli space (see figure 2).

Even without averaging, a Narain code CFT is related to an abelian Chern-Simons theory. Indeed (89) and (95) imply that the partition function is given as a finite sum involving $U(1)_{2p}$ characters $\Theta_{m,p}(\tau)/\eta(\tau)$ and is therefore a rational CFT with an extended chiral algebra corresponding to the $U(1)_{2p}^n$ Chern-Simons theory. (See for example [61].) It would be interesting to see if the conjectural holographic description above can be obtained from an ensemble of Chern-Simons theories.

There are also other directions of research related to quantum codes and CFTs [62–68]. It deserves further investigation to see whether our construction is relevant to these recent developments.

---

[5]This statement follows from the direct calculation or Tauberian theorem (see, for example, Theorem 15.3 of section 1 in [59]).

[6]Note that we are focused on the large-$n$ limit of the averaged theory here.

## Acknowledgments

We are grateful to S. Yahagi for valuable discussions. The work of T. N. was supported in part by the JSPS Grant-in-Aid for Scientific Research (C) No.19K03863, Grant-in-Aid for Scientific Research (A) No. 21H04469, and Grant-in-Aid for Transformative Research Areas (A) "Extreme Universe" No. 21H05182 and No. 21H05190. The research of T. O. was supported in part by Grant-in-Aid for Transformative Research Areas (A) "Extreme Universe" No. 21H05190. The work of K. K. was supported by Forefront Physics and Mathematics Program to Drive Transformation (FoPM), a World-leading Innovative Graduate Study (WINGS) Program, the University of Tokyo.

## A   List of notations

| Symbol | Definition | See |
| --- | --- | --- |
| $p$ | A prime number. | |
| $\mathbb{F}_q$ | The field of order $q$. | |
| $H_p$ | Hilbert space of a qudit system with $p$ states . | |
| $\omega_p$ | The primitive $p$-th root of unity ($\omega_p = e^{2\pi i/p}$). | |
| $\mathsf{g}(\alpha,\beta)$ | The generalized Pauli operator on the single-qudit system. | Eq.(10) |
| $g(\alpha,\beta)$ | The generalized Pauli operator on the $n$-qudit system. | Eq.(13) |
| $\mathcal{P}_n^{(p)}$ | The $n$-qudit Pauli group. | |
| $\langle\cdot,\cdot\rangle$ | The symplectic bilinear form on $\mathbb{F}_p^{2n}$. | |
| $S$ | A stabilizer group. | |
| $V_S$ | The code subspace stabilized by a stabilizer group $S$. | |
| $N(G)$ | The normalizer of a subgroup $G$ in an appropriate group. | |
| $\mathsf{H}$ | The check matrix of a stabilizer code. | Eq.(20) |
| $\mathsf{G}$ | The generator matrix of a stabilizer code. | Eq.(25) |
| $\mathsf{W}$ | A matrix that defines the symplectic product $\langle\cdot,\cdot\rangle$ on $\mathbb{F}_p^{2n}$. | Eq.(22) |
| $I_n$ | The $n \times n$ identity matrix. | |
| $\mathsf{U}(n)$ | The unitary group of degree $n$. | |
| $C$ | A classical code on $\mathbb{F}_p^n$. | |
| $G_C$ | The generator matrix of a classical code $C$. | |
| $H_C$ | The parity check matrix of a classical code $C$. | |
| $c$ | A codeword of a classical code (written as a row vector on $\mathbb{F}_p^n$). | |
| $\cdot$ | The Euclidean inner product over $\mathbb{F}_p^n$. | |
| $C^{\perp}$ | The dual code of a classical code $C$ with respect to the Euclidean inner product. | Eq.(35) |
| $\mathsf{H}_{(C_X,C_Z)}$ | The check matrix of the CSS code constructed from $C_X$ and $C_Z$. | Eq.(38) |
| $G_{\mathsf{H}}$ | The generator matrix of the classical code with a check matrix $\mathsf{H}$. | Eq.(47) |
| $\mathcal{C}$ | The classical code generated by the matrix $G_{\mathsf{H}}$. | Eq.(48) |

| | | |
|---|---|---|
| $\eta$ | The off-diagonal Lorentzian metric. | Eq.(49) |
| $\odot$ | The inner product with respect to the metric $\eta$. | |
| $\mathcal{C}^{\perp}$ | The dual code of a classical code $\mathcal{C}$ with respect to the metric $\eta$. | Eq.(51) |
| $\Lambda(\mathcal{C})$ | The Construction A lattice from a classical code $\mathcal{C}$. | Eq.(54) |
| $\Lambda^{*}$ | The dual lattice of a lattice $\Lambda$ with respect to the metric $\eta$. | Eq.(55) |
| $\lambda$ | A lattice vector written as a row vector. | |
| $\widetilde{\eta}$ | The diagonal Lorentzian metric. | Eq.(74) |
| $\circ$ | The inner product with respect to the metric $\widetilde{\eta}$. | Eq.(73) |
| $\widetilde{\Lambda}(\mathcal{C})$ | The momentum lattice obtained by a linear transformation from the Construction A lattice $\Lambda(\mathcal{C})$. | |
| $(p_L, p_R)$ | A momentum vector that is an element of a momentum lattice. | |
| $Z_{\mathcal{C}}$ | The partition function of a Narain code CFT. | Eq.(79) |
| $\Theta_{\widetilde{\Lambda}(\mathcal{C})}$ | The lattice theta function of the momentum lattice obtained from a classical code $\mathcal{C}$. | Eq.(80) |
| $W_{\mathcal{C}}$ | The complete enumerator polynomial of a classical code $\mathcal{C}$. | Eq.(83) |
| $W_{C,C^{\perp}}^{(\mathrm{CSS})}$ | The complete enumerator polynomial of a classical code based on a CSS code with a check matrix $\mathsf{H}_{(C,C^{\perp})}$. | Eq.(105) |
| $\underline{C}$ | The product of $r$ classical codes: $\underline{C} = C^{(1)} \times \cdots \times C^{(r)}$ | |
| $\mathcal{W}_{\underline{C}}$ | The $r$-fold complete joint weight enumerator for $\underline{C} = C^{(1)} \times \cdots \times C^{(r)}$. | Eq.(107) |
| $Z_{C,C^{\perp}}^{(\mathrm{CSS})}$ | The partition function of a Narain code CFT based on a CSS code with a check matrix $\mathsf{H}_{(C,C^{\perp})}$. | |
| $W_{C,C}^{(\mathrm{CSS})}$ | The complete enumerator polynomial of a classical code based on a CSS code with a check matrix $\mathsf{H}_{(C,C)}$ for a self-dual code $C$. | Eq.(118) |
| $W_{g,C}$ | The genus-$g$ weight enumerator of a classical code $C$. | Eq.(120) |
| $\mathcal{M}_{n,p}$ | The set of classical self-dual codes $C \subset \mathbb{F}_p^n$. | Eq.(131) |
| $E_{n,p}^{(g)}$ | The average of genus-$g$ weight enumerators over the set $\mathcal{M}_{n,p}$. | Eq.(132) |
| $A$ | A tuple of non-negative integers $e_v$ where $v \in \mathbb{F}_p^g$. | |
| $\dim_2(A)$ | The dimension of a tuple $A$ for $p = 2$. | Eq.(133) |
| $\dim_p(A)$ | The dimension of a tuple $A$ for odd prime $p$. | Eq.(155) |
| $\mathfrak{c}$ | A tuple of $g$ codewords: $\mathfrak{c} = (c^{(1)}, \cdots, c^{(g)})$. | |
| $\mathfrak{C}$ | The classical code generated by $\mathbf{1}_n$ and $\mathfrak{c}$ for $p = 2$ and by $\mathfrak{c}$ for odd prime $p$. | |
| $\overline{W}_{n,p}^{(\mathrm{CSS})}$ | The averaged complete enumerator polynomial of CSS codes over self-dual codes $C \in \mathcal{M}_{n,p}$. | Eq.(168) |
| $\overline{Z}_{n,p}^{(\mathrm{CSS})}$ | The averaged partition function of Narain code CFTs based on a class of CSS codes. | |

# B  Saddle point computation

In this appendix we perform a saddle point computation of the integral (172) to derive the result (173). For this purpose, let us introduce the function

$$f(z) := \sum_{a,b} z_{ab} \log \frac{z_{ab}}{x_{ab}} . \tag{B.1}$$

We treat $z_{ab}$ with $(a,b) \neq (0,0)$ as independent variables. Using the relation $z_{00} = 1 - \sum_{(a,b)\neq(0,0)} z_{ab}$, we find for $(a,b) \neq (0,0)$ and $(c,d) \neq (0,0)$

$$\frac{\partial f}{\partial z_{ab}} = \log\left(\frac{z_{ab}}{z_{00}} \frac{x_{00}}{x_{ab}}\right), \qquad H_{ab;cd} := \frac{\partial^2 f}{\partial z_{ab} \partial z_{cd}} = \frac{\delta_{ac}\delta_{bd}}{z_{ab}} + \frac{1}{z_{00}} . \tag{B.2}$$

The saddle point $z_*$ defined as the solution of $\partial f / \partial z_{ab} = 0$ is

$$z_{*ab} = \frac{x_{ab}}{\sum_{c,d\in\mathbb{F}_p} x_{cd}} . \tag{B.3}$$

After several non-trivial cancellations in the saddle point computation of the integral, we are left with

$$\overline{W}_{n,p}^{(\text{CSS})}(\{x_{ab}\}) = p^{-n}\left(\sum_{a,b} x_{ab}\right)^n \left(\prod_{(a,b)} z_{*ab}\right)^{-1/2} (\det H|_{z=z_*})^{-1/2}\left(1 + \mathcal{O}(n^{-1})\right). \tag{B.4}$$

The Hessian matrix $H$ given in (B.2) is of the form

$$\text{diag}\left(\frac{1}{y_1}, \ldots, \frac{1}{y_L}\right) + \frac{1}{x}\begin{pmatrix}1\\ \vdots \\ 1\end{pmatrix}\begin{pmatrix}1 & \ldots & 1\end{pmatrix} = \frac{1}{x}\text{diag}\left(\frac{1}{y_1}, \ldots, \frac{1}{y_L}\right)(xI + B), \tag{B.5}$$

where

$$B = \begin{pmatrix}y_1\\ \vdots \\ y_L\end{pmatrix}\begin{pmatrix}1 & \ldots & 1\end{pmatrix} . \tag{B.6}$$

The determinant of $xI + B$ is the characteristic polynomial of $-B$, which is given by $x^{L-1}(x+y_1+\ldots+y_L)$ because the eigenvalues of $B$ are $0$ with multiplicity $L-1$ and $y_1+\ldots+y_L$. Then we find

$$\det H|_{z=z_*} = \left(\prod_{(a,b)} z_{*ab}\right)^{-1} . \tag{B.7}$$

Thus the third and the fourth factors in (B.4) exactly cancel out, giving the result (173).

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
