# Peer review of "Narain CFTs from qudit stabilizer codes"

_SciPost Physics Core, doi:SciPost Phys. Core 6, 035 (2023)_

## Round 1 · Referee Report · Anonymous (Referee 1) · 2023-1-23

Strengths

The paper builds on the recently-introduced relation between certain quantum error-correcting codes and 2d Narain CFTs by expanding the family of quantum codes with the CFT counterparts. It is a timely development and include the explicit results.

Weaknesses

A possible weakness is the limited scope - it would be interesting to ask the question in full generality: to outline all quantum codes with the CFT counterparts (of course this question can be asked only within a given framework, which by itself might be generalized in the future).

Report

The paper develops the connection between quantum codes and 2d CFTs, following the original construction of Refs. [1-3] for classical codes, and its extension to quantum codes in [4]. The latter was extended in Ref. [11] and then in Ref. [28], but the codes considered in these papers were classical. This prompts a natural question if quantum codes have any role in this case. The authors answer this question, at least partially, by constructing an explicit connection between a class of quantum error-correcting codes and certain classical codes discussed in [11]. The paper contains many explicit results, including corresponding CFT torus partition functions, as well as the ensemble-averaged partition function.

There are several natural questions which follow from this work. One if quantum codes considered by the authors have any physical interpretation in terms of CFT Hilbert space, in the spirit of Ref. [62]. Another question is to consider more general family of quantum codes giving rise to different types of CFT-related classical codes introduced in Ref. [28].

---

## Editorial Decision

published